# A quieter state of charge and ultra-low-noise of the collective current in quasi-1D charge-density-wave nanowires

Subhajit Ghosh [1,2], Nicholas Sesing [3], Zahra Ebrahim Nataj [1,2], Tina Salguero [3], Sergey Rumyantsev [4], Roger K. Lake[5] & Alexander A. Balandin [1,2,6] ✉

Electronic flicker noise limits phase stability in communication systems, reduces the sensitivity and selectivity of sensors, and degrades coherence in quantum devices. There is a strong need for unconventional materials and strategies for achieving ultra-low-noise performance in nanoscale and quantum electronics. Here, we demonstrate that in nanowires of the quasi-one-dimensional, fully gapped charge-density-wave material $(TaSe_4)_2I$, low-frequency electronic noise is suppressed below the limit of thermalized charge carriers in passive resistors. When the current is dominated by the sliding Frohlich condensate, the normalized noise spectral density, $S_I/I^2$, decreases linearly with current, $I$ — a striking departure from the constant value of $S_I/I^2$, observed in conventional conductors. No residual minimum noise level is reached for the current of the electron-lattice condensate in $(TaSe_4)_2I$ nanowires. Repeating the measurements for another charge-density wave conductor, $NbS_3$-II, we found a similar reduction below the normal electron limit at room temperature. Our findings signal intrinsically lower current fluctuations within a correlated electron transport regime.

Recent years have witnessed renewed interest in charge density waves (CDW)[1–4] driven by the exciting physics of strongly correlated phenomena, the growing number of quasi-one-dimensional (1D) and quasi-two-dimensional (2D) van der Waals materials with CDW phases, and possibilities for practical applications[5–13]. In an ideal 1D system, a collective transport mode of the CDW condensate can slide without dissipation as the Frohlich current[14]; in real materials, however, due to pinning by defects and surface imperfections, a finite electric field exceeding the threshold field, $E_t$, is required to depin the CDW and initiate sliding[1–4]. The CDW sliding results in the collective, electron-lattice condensate contribution to the total current, evident from the appearance of an AC component under DC bias, Shapiro-like steps in the current with applied RF signal, and an overall nonlinear increase in

current[1–4]. While achieving the dissipation-less Frohlich current[14] of the sliding electron–lattice condensate appears to be impossible in real materials, one can imagine an important related target, namely reaching the electron transport regime where electronic noise is inhibited due to the collective, strongly-correlated nature of the electron-lattice condensate current.

We address here the fundamental questions in the physics of electron transport: is the collective current of the sliding, strongly-correlated CDW condensate less noisy than that of the normal current? Can we inhibit the low-frequency electronic noise in CDW conductors below the noise limit of normal electrons in conventional conductors? The low-frequency noise in electronic materials, also known as excess or flicker noise, has the current spectral density, $S_I$, scaling as $1/f$, and,

[1]Department of Materials Science and Engineering, University of California, Los Angeles, Los Angeles, CA, USA. [2]California NanoSystems Institute, University of California, Los Angeles, Los Angeles, CA, USA. [3]Department of Chemistry, University of Georgia, Athens, GA, USA. [4]Institute of High-Pressure Physics, Polish Academy of Sciences, Warsaw, Poland. [5]Department of Electrical and Computer Engineering, University of California, Riverside, Riverside, CA, USA. [6]Center for Quantum Science and Engineering, University of California, Los Angeles, Los Angeles, CA, USA. ✉e-mail: balandin@seas.ucla.edu

in some cases, it reveals Lorentzian bulges superimposed on the $1/f$ background ($f$ is the frequency). In the CDW context, it has been termed as "broadband noise"[15–17]. The importance of this type of noise is well-known[18]. It up-converts via the device's non-linearities and appears in high-frequency signals as phase noise, thus degrading the communication systems. It sets the fundamental limit to the sensitivity and selectivity of any sensor because its power increases with the measurement time, owing to the $1/f$ spectral density[19,20]. Continuous downscaling of electronic technologies leads to an increasing relative level of $1/f$ noise with decreasing device area and thus numerous problems with noise control[21]. Reducing this noise type is crucial in quantum computing technologies because it degrades qubit coherence times and limits quantum sensors[22–26].

We hypothesized that the current carried by a sliding CDW strongly correlated electron-lattice condensate is inherently less noisy than that of thermalized single electrons. Accordingly, when transport is dominated by the condensate, the noise is suppressed relative to the noise level of the current of drifting, uncorrelated, thermally excited electrons. To test this, we focused on CDW systems in the incommensurate phase (IC-CDW), where the density of normal carriers is low and the current is dominated by the collective CDW contribution. Noise suppression is expected to be most pronounced at bias voltages above the depinning threshold, since at the depinning point, CDW creep and nonuniform sliding generate excess fluctuations[27,28]. Upon considering various quasi-1D CDW van der Waals materials[29], we selected two candidates: a Weyl CDW semimetal $(TaSe_4)_2I$[11,30,31] and the monoclinic CDW polymorph, $NbS_3$-II[32,33]. The first material, $(TaSe_4)_2I$, undergoes a Peierls-type transition to the IC-CDW phase, accompanied by an opening of a relatively large energy gap, at a temperature $T_P$ in the range of 245 K − 260 K. The second material, $NbS_3$-II, reveals three CDW phases at Peierls temperatures $T_{P0}$ = 460 K, $T_{P1}$ = 330 K to 370 K, and $T_{P2}$ = 150 K[32]. The fact that $NbS_3$-II exhibits a CDW electron-lattice condensate phase above room temperature (RT) makes this material promising for practical applications. Using these two quasi-1D materials, we fabricated nanowire-geometry test structures to promote coherent CDW sliding[1–4].

In this work, we demonstrate that the low-frequency noise in CDW nanowire conductors can drop below the noise limit of normal electrons. In both $(TaSe_4)_2I$ and $NbS_3$-II, the total normalized noise is lower than that of individual electrons. Our experiments with $(TaSe_4)_2I$ found no residual minimum noise level at higher biases, suggesting minimal or no intrinsic noise from the sliding electron-lattice condensate. Therefore, the intrinsic noise of the sliding condensate is negligible compared to the noise of the normal electrons or the fluctuations in the threshold field. The noise due to fluctuations at the depinning threshold is extrinsic and caused by lattice imperfections that locally pin the condensate. Once the bias voltage is well past threshold and the sliding mode is established, the total normalized noise drops below the noise of normal electrons. In the case of $NbS_3$-II, the noise of the CDW condensate appears to emerge at higher biases, saturating the overall noise at a level lower than that of the normal electron noise. The total normalized noise spectral density in CDW conductors decreases with current, a striking difference from conventional metals or semiconductors, where the normalized noise level remains constant with the current. The noise reduction at the higher current densities gives an advantage to CDW conductors in signal-to-noise ratios. These insights into the fundamental physics of CDW nanowires could be transformative for quantum technologies and suggest a way for future noiseless electronics.

## Results
### Electrical current of the sliding CDW condensate in $(TaSe_4)_2I$ nanowires
First, we confirmed that our $(TaSe_4)_2I$ device structures are of high quality and support a fully-gapped CDW transport regime. Figure 1a

shows scanning electron microscopy (SEM) images and energy-dispersive spectroscopy (EDS) maps of the synthesized material, revealing the ribbon-like structure of the quasi-1D van der Waals crystal and uniform elemental distribution. Figure 1b is an optical microscopy image of one of the four-contact nanowire structures with a channel thickness of ~83 nm. Figure 1c shows the channel resistance normalized by the RT resistance as a function of inverse temperature, $1000/T$, measured in the linear regime at a small bias. The peak in the resistance slope is associated with the material's transition from the normal metal to IC-CDW phase at $T_P \approx 246$ K. Below $T_P$, an opening of the Peierls bandgap in the Fermi surface is accompanied by a change in the resistance, described as $\sigma/\sigma_0 = \exp(-\Delta/K_B T)$, where $\sigma$ is the conductivity, $\sigma_0$ is the conductivity at RT, $K_B$ is the Boltzmann constant, and $\Delta$ is the activation energy. For the device in Fig. 1b-f, we extracted the bandgap, $E_g = 2\Delta \approx 300$ meV, in agreement with the literature[34,35]. To further confirm the IC-CDW phase, Fig. 1d shows the differential resistance as a function of bias voltage for temperatures between 150 K and 220 K. This dependence was obtained by numerical differentiation of measured I–Vs. The nearly constant differential resistance at low bias corresponds to the Ohmic resistance of the normal charge carriers. At the threshold bias voltage, $V_t$, the resistance decreases due to the onset of CDW sliding. The $V_t$ value shows a non-monotonic temperature dependence consistent with trends reported for other quasi-1D CDW systems[1,3,4]. Slight deviations from the perfectly flat derivative characteristics at low bias and kinks are attributed to CDW creep before the depinning and not completely coherent CDW depinning and sliding.

The I–Vs and differential conductance, $G$, for the $(TaSe_4)_2I$ nanowire are presented in Fig. 1e, f, respectively. These characteristics are measured for the second time during the noise measurements. They are further used in the noise data analysis. The differential conductance includes the normal and collective current components, $G = G_n + G_c$. The onset of CDW sliding is seen at $V_t \approx 0.08$ V, when the total current becomes super-linear, $I \propto V^\alpha$ ($\alpha$ = 1.5–2.0). The nonlinear increase in total current is due to the contribution of the CDW condensate. The collective component of the current, $I_c$, and conductance, $G_c$, are shown by the blue circles in panels (e) and (f), respectively. The data in Fig. 1a–f prove that we have high-quality CDW nanowire conductors with the current dominated by sliding electron-lattice condensate at the bias of ~1 V.

### The electronic noise of the CDW condensate in $(TaSe_4)_2I$ nanowires
To probe noise behavior in $(TaSe_4)_2I$ devices in the CDW regime, we performed spectral measurements below the Peierls transition temperature $T_P$, ensuring the system remained in the IC-CDW phase. Figure 2a–d show the normalized current spectral density, $S_I/I^2$, as a function of frequency, $f$, for four different bias regions: (I) low-bias–purely normal carrier transport; (II) near the CDW depinning threshold; (III) at the onset of CDW sliding; and (IV) high bias, where the sliding condensate dominates. Across all regions, the noise exhibits $1/f$ spectral behavior. The Lorentzian bulges appearing near the CDW depinning threshold in regions II and III are a signature of phase transitions or depinning[36,37]. Figure 2e shows the noise, $S_I/I^2$, at a fixed frequency $f$ = 10 Hz, as a function of bias voltage, across all four regions. The noise peaks near the depinning voltage, $V_t$, where the CDW starts to slide. As the bias voltage increases above $V_t$, the noise decreases, approaching the level of normal electrons. However, at the bias voltage of ~0.7 V, where the current is dominated by the sliding CDW condensate, we observe a striking feature: instead of saturating at the noise level of electrons in a passive conductor (region I), the total noise level decreases below this limit (yellow shade part of region IV). The noise in this fully-gapped quasi-1D CDW nanowire is, therefore, inhibited below the normal metal

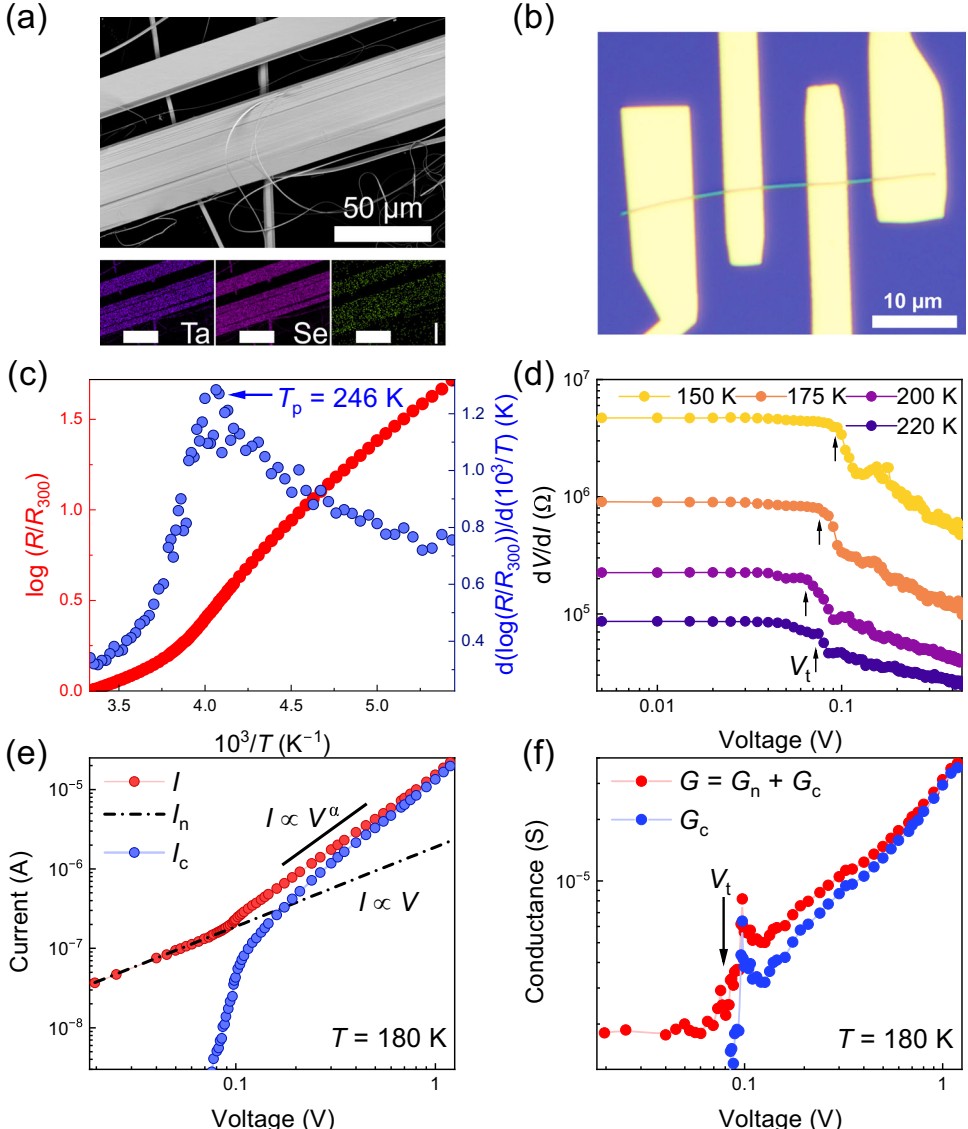

**Fig. 1 | Transport characteristics of (TaSe$_4$)$_2$I nanowires. a** SEM and EDS maps of synthesized (TaSe$_4$)$_2$I. In the EDS mapping, the elements Ta, Se, and I are represented in purple, magenta, and green colors, respectively. The scale bar is 50 μm. In the images, white/dark colors correspond to high/low electron emission. **b** Optical microscopy image of a (TaSe$_4$)$_2$I nanowire structure with several top metal electrodes. The white/dark colors correspond to high/low light reflection. **c** Temperature dependence of the resistance, showing the CDW transition at $T_P \approx$ 246 K. **d** Differential resistance of the (TaSe$_4$)$_2$I nanowire, revealing the threshold voltage, $V_t$, of the CDW depinning. **e** I–V characteristics of (TaSe$_4$)$_2$I nanowire in the incommensurate CDW phase. Note the onset of CDW condensate sliding at $V_t \approx 0.08$ V. **f** The conductance dependence on the applied bias at $T = 180$ K. The collective current component of the CDW sliding condensate is shown with blue circles in (**e**, **f**). The data are for device 1.

limit when the current is dominated by the sliding electron-lattice condensate. This behavior was consistently observed across multiple devices (Fig. 2f and Supplementary Materials).

Figure 3a–f illustrates the evolution of the noise spectra in the bias region II, near the CDW depinning threshold, indicating the association of the Lorentzian features with the depinning. The Lorentzian noise component diminishes as the bias voltage surpasses the depinning threshold and enters the CDW sliding regime. The observed Lorentzian bulges originate from threshold fluctuations; they are associated with the system fluctuations between the pinned and depinned states. In mathematical description, the transitions between two phases in the material are similar to those in the two-level systems, which result in the Lorentzian features in the noise spectra. Additional data is available in the Supplementary Materials.

## Analysis of the experimental noise data in (TaSe$_4$)$_2$I nanowires

The noise reduction in the (TaSe$_4$)$_2$I nanowires is intriguing and counterintuitive, thus requiring theoretical explanation. In a simple approach, the fluctuations of the conductance of the normal and collective currents can be described by the noise spectral density, $S_I/I^2$:

$$\frac{S_I}{I^2} = \frac{S_G}{G^2} = \frac{S_{G_n}}{G_n^2}\frac{G_n^2}{(G_n + G_c)^2} + \frac{S_{G_c}}{G_c^2}\frac{G_c^2}{(G_n + G_c)^2}. \quad (1)$$

Here, the conventional assumption is that the fluctuations in the two conductivities are independent and the cross-correlation can be neglected. At low bias, $V < V_t$, the contribution of the CDW to the current is zero, $G_c = 0$, and one can determine the normal conductance noise, $S_{G_n}/G_n^2$, from the experimental data. The relative contributions of

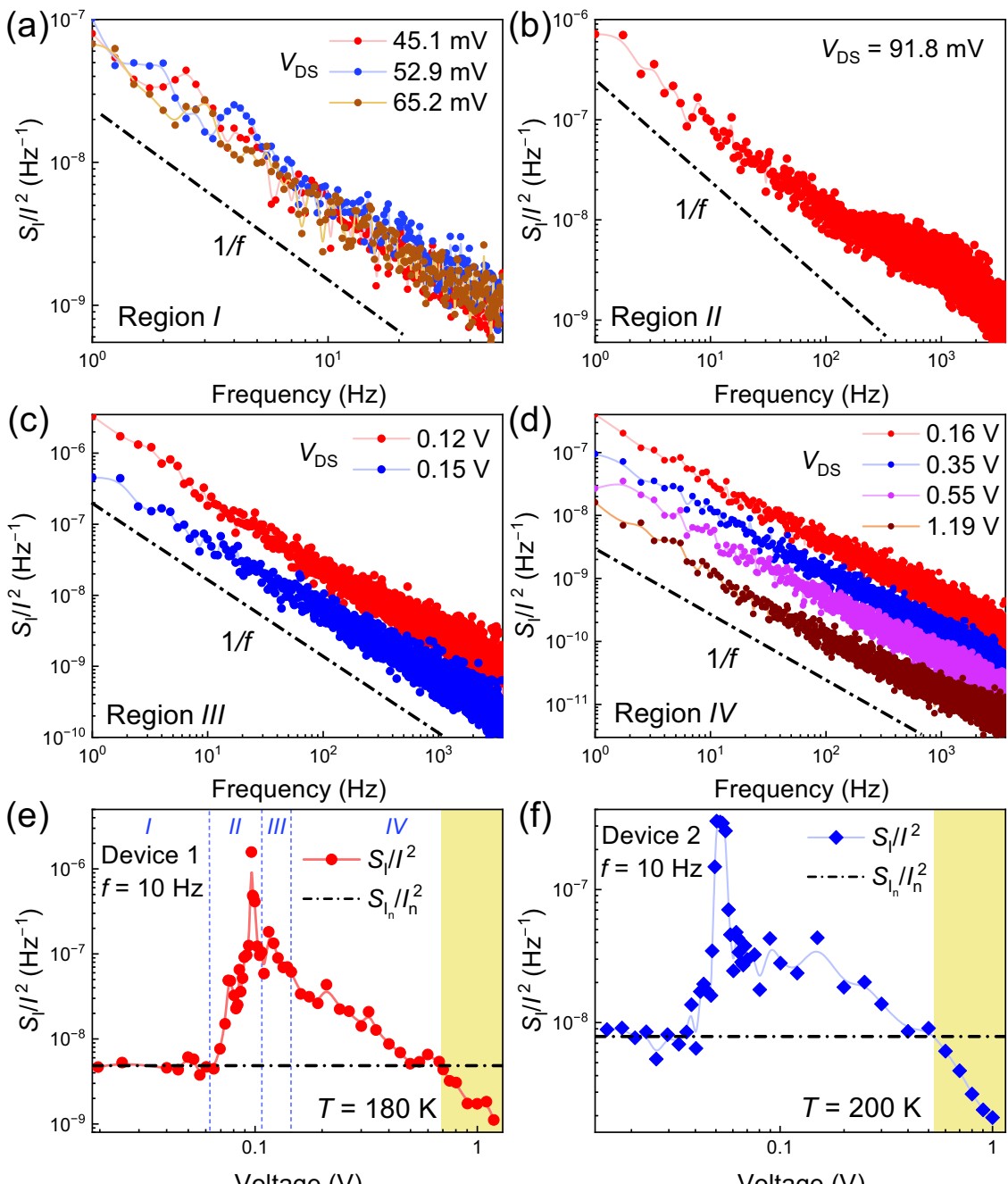

**Fig. 2 | Electronic noise inhibition in (TaSe$_4$)$_2$I nanowires. a** The normalized noise spectral density, $S_I/I^2$, in the linear regime at low bias (region I). The noise spectrum is of the $1/f$ type characteristic for metals and semiconductors. **b** $S_I/I^2$ at the CDW depinning point (region II). Note the emergence of Lorentzian bulges over the $1/f$ envelope. **c** $S_I/I^2$ at the onset of CDW condensate sliding (region III). **d** $S_I/I^2$ at higher biases when the collective current of the sliding CDW condensate becomes dominant (region IV). **e** The noise $S_I/I^2$, at fixed frequency $f = 10$ Hz vs. bias voltage, measured at $T = 180$ K. The noise of the CDW collective current (yellow area of region IV; bias above 0.7 V) drops below the noise limit of the individual charge carriers (region I; dashed line represents the average noise of individual carriers). **f** The same as in the (**e**) for a different device at $T = 200$ K. The data are shown for device 1 in (**a**–**e**) and device 2 in (**f**).

the normal, $[G_n/(G_n + G_c)]^2$, and the collective, $[G_c/(G_n + G_c)]^2$, components of electrical conductivity are also taken from the experiment (Fig. 4a). The nonlinear CDW conductance scales as $G_c \propto I^\beta$, where $\beta \approx 0.5$ (Fig. 4b), and no saturation is observed. Figure 4c presents the weighted noise contributions of the normal and CDW currents to the overall noise. At low bias, all noise is due to the normal current of individual electrons, $S_{G_n}/G_n^2$, independent of voltage (horizontal line, gray symbols). At high current, all noise is due to the CDW current, $S_I/I^2 \approx S_{G_c}/G_c^2[G_c/(G_n + G_c)]^2 \approx S_{G_c}/G_c^2$ (red). The noise from fluctuations of the normal conductance remains the same, but its contribution

to the total noise, $S_{G_n}/G_n^2[G_n/(G_n + G_c)]^2$ (blue) decreases because it is shunted by increasing CDW conductance. The shunting of the noise of normal electrons occurs because of the dominance of the CDW conductance (Fig. 4a) and the intrinsically lower noise of sliding CDWs. A simple Eq. (1) can be extended to include the fluctuations in the threshold voltage. We have accomplished this task using a conventional linear model approximation for the current. The details are provided in the Methods section. The noise model in the linear approximation confirms the principal possibility of reducing the total normalized noise below the normal electron limit. However, this model does not describe

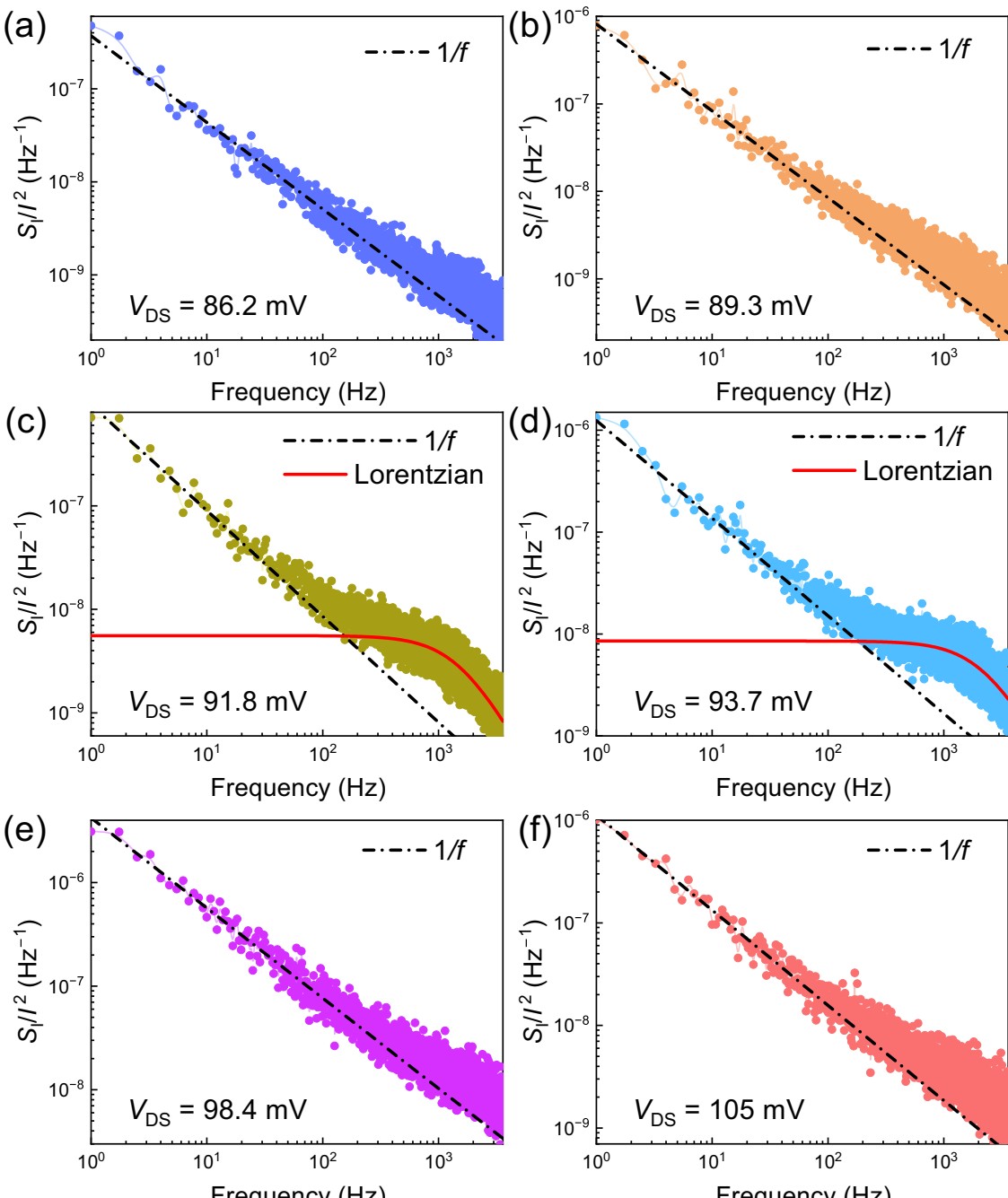

**Fig. 3 | Details of the noise characteristics near depinning for (TaSe₄)₂I nanowires. a–f** The noise spectra at different voltages across the threshold field $V_t$ (region II). The noise exhibits a $1/f$ spectrum, characteristic of flicker noise, with Lorentzian bulges superimposed on the $1/f$ envelope at intermediate bias points, where the noise peak appears. The emergence of Lorentzian bulges is common at the phase transition points, including at CDW depinning. The data are shown for device 1.

all noise features accurately since the CDW current is strongly nonlinear. The accurate noise description requires a comprehensive phenomenological theory, which is presented in the next section.

The noise reduction below the normal resistor limit is not the only surprise feature observed in experimental data. Figure 4d shows the total normalized noise spectral density, $S_I/I^2$, as a function of current, I, in the CDW sliding regime. The noise reduces with increasing current as $S_I/I^2 \propto 1/I$. Such inverse scaling with current contrasts drastically with the noise dependence on current for linear passive resistors, where the noise level does not depend on current: $S_I/I^2 \propto constant$ (or $S_I \propto I^2$)[38]. The fact that we observed a different dependence on the normalized

noise spectral density in a CDW conducting channel is of great importance. It is well-known that the noise level, i.e., normalized noise spectral density, in passive resistors does not depend on current: $S_I/I^2 \propto$ constant (or $S_I I^2$). This fundamental property originates from the linear response theory[38]. In terms of physics, it means that the current does not drive the resistance fluctuations in the sample but rather makes them visible via Ohm's law. The fact that $S_I I^2$ allowed for the introduction of the phenomenological Hooge formula, $S_I/I^2 \sim \alpha_H/fN$, where $\alpha_H$ is the Hooge parameter, and $N$ is the number of charge carriers in the resistor. There are known cases of deviation from $S_I/I^2 \propto$ constant dependence in specific electronic devices, like diodes[39]. In this work, we

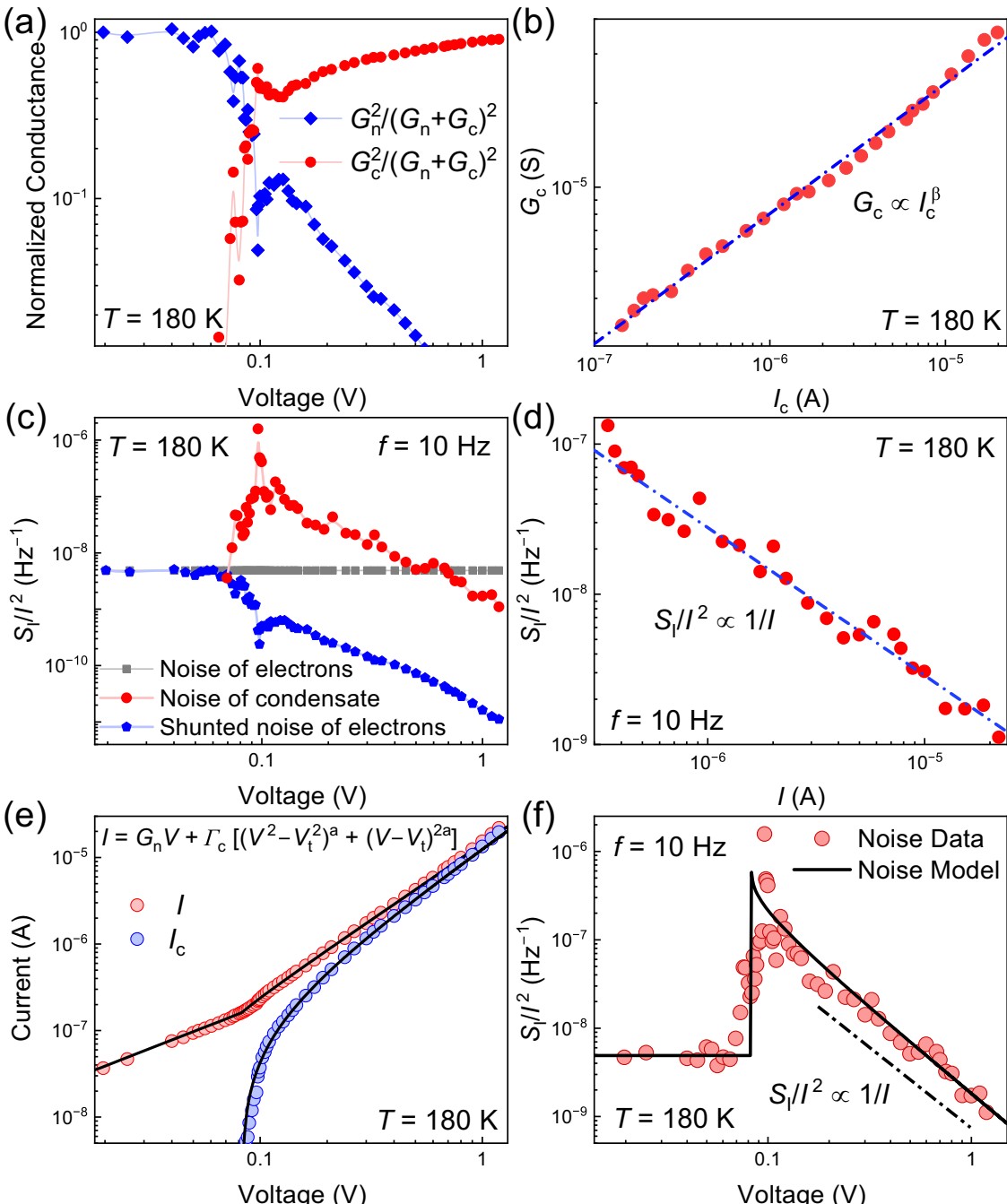

**Fig. 4 | Physical mechanism of noise inhibition in (TaSe$_4$)$_2$I nanowires. a** The relative conductance contribution of normal, $[G_n/(G_n + G_c)]^2$, and CDW carriers, $[G_c/(G_n + G_c)]^2$, as a function of bias voltage. **b** CDW conductance, $G_c$ dependence on CDW current $I_c$. The blue dashed line shows a power-law fit $G_c \propto I_c^\beta$ to the data on the log-log scale. **c** Noise of electrons (gray symbols), weighted noise of electrons (blue), and weighted noise of the sliding CDW condensate (red) as a function of bias voltage. **d** $S_I/I^2$, as a function of current. Note that the normalized noise scales inversely with the current, unlike passive resistors, which maintain a constant noise level. The blue dashed line shows a power-law fit $S_I/I^2 \propto 1/I$ to the data on the log-log scale. **e** Measured I–Vs fitted with our model based on Eq. (2). **f** Noise calculated from Eq. (3) with the experimental data superimposed. No saturation is observed for the total normalized noise level in (TaSe$_4$)$_2$I nanowires, indicating that we are not reaching the intrinsic CDW noise floor. Details for the theory in (**e**, **f**) are in the "Methods". The data are shown for device 1.

report the observation of as $S_I/I^2 \propto 1/I$ scaling of noise in a generic resistor with two metal contacts. This means that increasing the current density will simultaneously decrease the noise level, which can become a game-changing advantage for interconnect applications.

## The general phenomenological theory of noise in CDW condensate conductors

To further understand the unusual noise reduction in fully-gapped CDW nanowires, we developed a comprehensive phenomenological

noise theory in nonlinear approximation. It has been accomplished by extending the "two-fluid" approach to include explicitly the nonlinear voltage dependence of the collective current in (TaSe$_4$)$_2$I. The total current, $I = I_n + I_c$, consists of a normal current $I_n = G_n V$, and the sliding CDW current modeled as[40]:

$$I_c = I_{c_1} + I_{c_2} = \Gamma_c \left[ \left( V^2 - V_t^2 \right)^a + (V - V_t)^{2a} \right] \theta(V - V_t). \quad (2)$$

Here, $\theta(V - V_t)$ is the unit step function that ensures the CDW contribution begins only after depinning. Equation (2) provides a phenomenological description of the CDW contribution to the current in our nanowire devices. The two power-law terms originate from the standard CDW current expressions quoted by Grüner[40], but the exponents $a$ and $2a$ are employed as effective exponents that capture the voltage dependence of the sliding condensate over the full experimental voltage range. We further constrain the ratio of the exponents of the two nonlinear terms to share a single generalized conductance $\Gamma_c$ with fixed dimensionality, which keeps the parametrization minimal and avoids introducing additional prefactors. This yields an accurate minimal model that reproduces the full nonlinear I–V characteristics and forms the basis for the quantitative noise analysis below.

We denote the two components of the collective current with different nonlinear voltage dependencies as $I_{c_1} = \Gamma_c(V^2 - V_t^2)^a \theta(V - V_t)$ and $I_{c_2} = \Gamma_c(V - V_t)^{2a}\theta(V - V_t)$, and $\Gamma_c$ is a generalized conductance relating the current to the nonlinear voltage. The transport behavior of our device exhibits a distinct kink at $V_t$. Similar changes in the bias dependence near threshold were previously attributed to the CDW creep before the onset of condensate sliding[41–43]. Our phenomenological model incorporates two nonlinear terms with distinct power dependencies to account for the initial depinning dynamics and the subsequent sliding behavior of the CDW beyond the threshold. The accuracy of the description is confirmed by the excellent fit to the experimental data of all examined devices (Fig. 4e and Supplementary Materials).

Assuming uncorrelated, zero-mean fluctuations in the physical parameters, $G_n$, $\Gamma_c$, and $V_t$, the total first-order fluctuation in current can be expressed as:

which are then used as inputs to the noise model. The first observation is that our theory accurately describes the noise peak due to fluctuations in $V_t$ and how it rolls off as $1/I$. The bias dependence of the weighted normal electron noise and the CDW condensate noise are described as $[(\langle\delta G_n^2\rangle/G_n^2)(I_n^2/I^2)]$ and $[(\langle\delta\Gamma_c^2\rangle/\Gamma_c^2)(I_c^2/I^2)]$, respectively. The noise of the normal electrons dominates below the threshold voltage, $V_t$. Both noise components are constant at lower and higher bias regimes, when $I \approx I_n$ and $I \approx I_c$, respectively. In the high-bias regime where CDW noise emerges, it can saturate the total noise if its amplitudes grow large enough. The bias dependence of the threshold field fluctuations, $\langle\delta V_t^2\rangle/V_t^2$, becomes pronounced near $V_t$. The total contribution from the threshold fluctuations to the noise contains the three distinct components within the square brackets of Eq. (4), each with a characteristic bias dependence. The first one, $S_{t_1}/I^2 = \{4a^2V_t^4/(V^2 - V_t^2)^2\}(\frac{I_{c_1}^2}{I^2})(\frac{\langle\delta V_t^2\rangle}{V_t^2})$ dominates near threshold and produces a noise peak that rapidly drops as $1/I^2$. The second one, $S_{t_2}/I^2 = \{4a^2V_t^2/(V - V_t)^2\}(\frac{I_{c_2}^2}{I^2})(\frac{\langle\delta V_t^2\rangle}{V_t^2})$ dominates at $V \gg V_t$, resulting in noise scaling as $1/I$. The third one $S_{t_3}/I^2 = [8a^2V_t^3/\{(V^2 - V_t^2)(V - V_t)\}]\{\frac{(I_{c_1}I_{c_2})}{I^2}\}(\frac{\langle\delta V_t^2\rangle}{V_t^2})$ drops approximately as $1/I^{1.5}$ in the intermediate regime. Together, these components capture accurately the evolution of the noise across the depinning transition and into the CDW sliding regime. The details of the model fitting and resulting noise behavior across different bias regimes are presented in the Supplemental Materials.

Significantly, our general phenomenological theory and Eq. (4) resolve the half-century-old argument about the role of fluctuations in the threshold field on the overall "broadband noise" level of CDW

$$\delta I = \frac{\partial I}{\partial G_n}\delta G_n + \frac{\partial I}{\partial \Gamma_c}\delta\Gamma_c + \frac{\partial I}{\partial V_t}\delta V_t$$

$$= \frac{G_n V \delta G_n}{G_n} + \left\{\frac{\Gamma_c\left[\left(V^2 - V_t^2\right)^a + (V - V_t)^{2a}\right]\delta\Gamma_c}{\Gamma_c} - \frac{2a\Gamma_c V_t\left[V_t^2\left(V^2 - V_t^2\right)^{(a-1)} + V_t(V - V_t)^{(2a-1)}\right]\delta V_t}{V_t}\right\}\theta(V - V_t). \tag{3}$$

Dividing by the total current, squaring, and taking the ensemble average, we obtain the normalized noise power expressed as:

$$\frac{\langle\delta I^2\rangle}{I^2} = \frac{\langle\delta G_n^2\rangle}{G_n^2}\left(\frac{I_n}{I}\right)^2 + \left\{\frac{\langle\delta\Gamma_c^2\rangle}{\Gamma_c^2}\left(\frac{I_c}{I}\right)^2 + \frac{\langle\delta V_t^2\rangle}{V_t^2}\left[\frac{2a\Gamma_c\left\{V_t^2\left(V^2 - V_t^2\right)^{(a-1)} + V_t(V - V_t)^{(2a-1)}\right\}}{I}\right]^2\right\}\theta(V - V_t)$$

$$= \frac{\langle\delta G_n^2\rangle}{G_n^2}\frac{I_n^2}{I^2} + \left\{\frac{\langle\delta\Gamma_c^2\rangle}{\Gamma_c^2}\frac{I_c^2}{I^2} + 4a^2\left[\frac{V_t^4}{\left(V^2 - V_t^2\right)^2}\frac{I_{c_1}^2}{I^2} + \frac{V_t^2}{(V - V_t)^2}\frac{I_{c_2}^2}{I^2} + \frac{2V_t^3}{\left(V^2 - V_t^2\right)(V - V_t)}\frac{I_{c_1}I_{c_2}}{I^2}\right]\frac{\langle\delta V_t^2\rangle}{V_t^2}\right\}\theta(V - V_t). \tag{4}$$

The first term is the noise of the normal current, the second term is the noise due to fluctuations of the generalized CDW conductance, $\Gamma_c$, and the third term is the noise due to fluctuations in the threshold voltage, $V_t$. At large voltages ($V \gg V_t$), the terms proportional to $\langle\delta V_t^2\rangle/V_t^2$ are suppressed, and Eq. (4) reduces to a form similar to Eq. (1). At $V = V_t^+$, the normalized noise due to fluctuations in $V_t$ is singular for $a < 1$, but in the limit of large $V \gg V_t$, it falls off rapidly.

Let us look closer at Fig. 4f, which shows the noise calculated from Eq. (4) superimposed on the experimental data. The fitting to the experimental I–Vs (see Fig. 4e), allows one to extract the parameters,

materials[15–17]. The "threshold noise" is the dominant mechanism for typical voltages in the CDW sliding regime, studied previously for different materials. It is extrinsic, defect-related, rather than intrinsic to CDWs. The same lattice imperfections and defects that pin the CDW also give rise to stochastic variations in the depinning threshold, establishing a direct connection between material disorder and low-frequency noise. However, in our $(TaSe_4)_2I$ nanowires, the total noise reduces rapidly beyond $V_t$, not only returning to the noise limit before the CDW depinning but falling below it. This raises the most important question: how noisy is the sliding CDW condensate itself compared to the current of normal electrons?

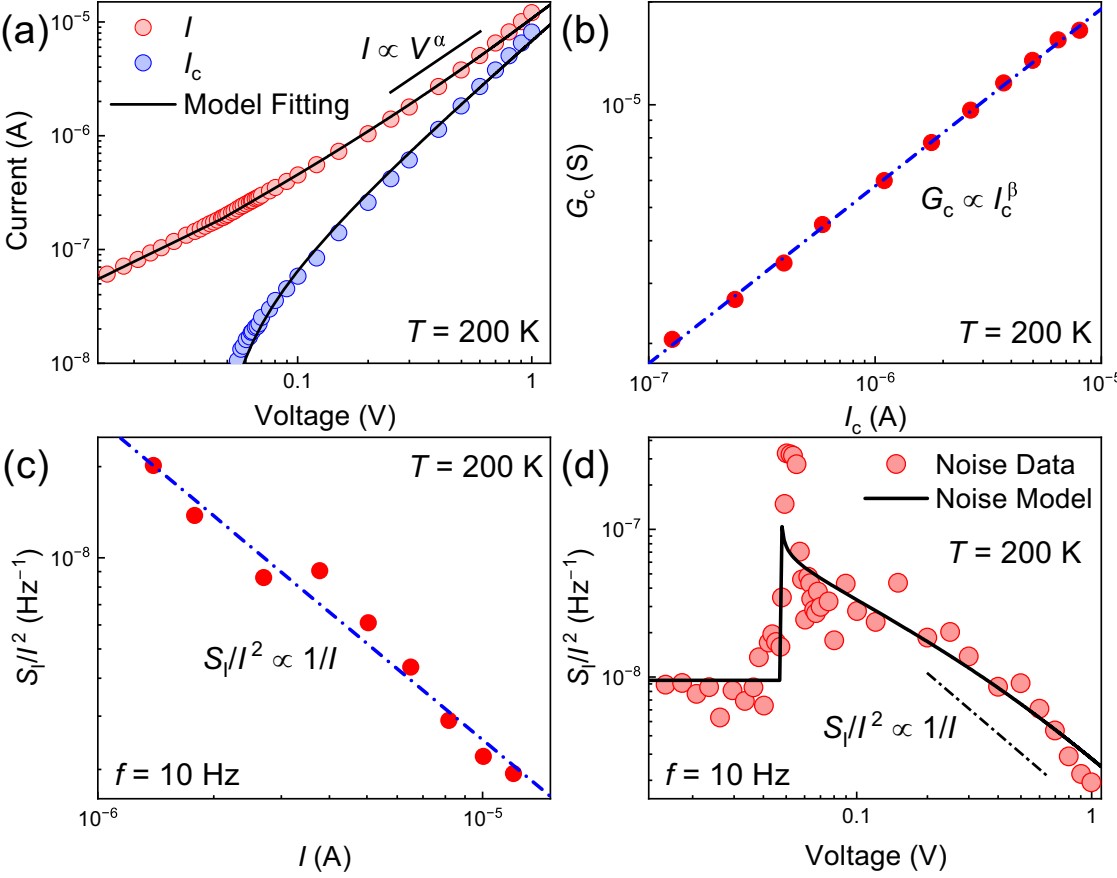

**Fig. 5 | Transport and noise behavior for a representative (TaSe$_4$)$_2$I nanowire device. a** The I–V characteristics at $T = 200$ K as fitted with the I–V model. **b** The CDW conductance, $G_c$, dependence on the CDW current, $i_c$, as $G_c \propto I_c^\beta$, where, $\beta = 0.5$. The blue dashed line shows a power-law fit $G_c \propto I_c^\beta$ to the data on the log–log scale. **c** $S_I/I^2$, as a function of current $I$, scales inversely with the current.

The blue dashed line shows a power-law fit $S_I/I^2 \propto 1/I$ to the data on the log–log scale. **d** The normalized noise spectral density calculated from the noise model, superimposed on the experimental noise data. The data are shown for device 2. Other (TaSe$_4$)$_2$I nanowire devices demonstrated similar behavior. The data for device 3 is provided in the Supplementary Information.

Our theory, via Eq. (4), establishes the relevant noise mechanisms and the noise scaling with the bias voltage. The noise amplitudes, $\langle\delta G_n^2\rangle/G_n^2$, $\langle\delta\Gamma_c^2\rangle/\Gamma_c^2$, and $\langle\delta V_t^2\rangle/V_t^2$ cannot be calculated from the first principles, but rather determined by comparing the experimental data with the model (see Supplementary Materials). Up to the maximum measured bias we only observe the noise due to fluctuations in $V_t$, and we do not observe any indication of a lower limit set by fluctuations in $\Gamma_c$. The normal and threshold noise amplitudes $\langle\delta G_n^2\rangle/G_n^2$ and $\langle\delta V_t^2\rangle/V_t^2$ can be determined from the fitting to experimental data as $4.9 \times 10^{-9}$ and $3.5 \times 10^{-7}$, respectively. The CDW noise amplitude, $\langle\delta\Gamma_c^2\rangle/\Gamma_c^2$, which can saturate the total noise level at high currents, is never revealed in our experiments, as seen clearly from the total noise, which continues falling below the normal electron limit (Fig. 2e, f). The upper bound on $\langle\delta\Gamma_c^2\rangle/\Gamma_c^2$ has to be taken as orders-of-magnitude smaller than that of $\langle\delta G_n^2\rangle/G_n^2$ and $\langle\delta V_t^2\rangle/V_t^2$ to obtain an accurate fit to the experiment. In the low-bias linear regime, the noise is constant; it originates from the normal carrier fluctuations described by $[(\langle\delta G_n^2\rangle/G_n^2)(I_n^2/I^2)]$. The prominent noise peak at the threshold voltage, $V_t$, arises from the $S_{t_1}/I^2$ component, while the gradual noise decay at higher biases follows a $1/I$ dependence and is governed by the $S_{t_2}/I^2$. The CDW noise, $[(\langle\delta\Gamma_c^2\rangle/\Gamma_c^2)(I_c^2/I^2)]$, is found to be insignificant across the entire measured voltage range and does not contribute to the observed noise. The total modeled noise, with all the noise components superimposed on the experimental data, is shown in the Supplemental Materials. To summarize, we found that the intrinsic noise of the sliding condensate in our (TaSe$_4$)$_2$I nanowires is negligible compared to the noise of normal electrons and the extrinsic noise of

the threshold-field fluctuations. This is consistent with the picture of the ideal sliding Fröhlich condensate as a dissipation-less, noise-free process, with all observed noise originating from defects and disorder.

## Reproducibility of the characteristics and absence of local heating effects

Given the fundamental nature of the noise reduction effect, we verified its reproducibility across multiple test structures. In Fig. 5a–d, we present the transport and noise characteristics of a second representative (TaSe$_4$)$_2$I nanowire device. The data for the third (TaSe$_4$)$_2$I nanowire test structure are included in the Supplemental Information. In all cases, we observed a consistent nonlinear conduction scaling as, $G_c \propto I_c^\beta$ (Fig. 5b) and noise reduction in the high-bias sliding regime following the $S_I/I^2 \propto 1/I$ law (Fig. 5c), which indicates that the noise reduction below the normal electron limit with no observed noise saturation is a robust and reproducible noise feature intrinsic to this CDW material.

In the data analysis, it is also important to exclude local heating effects. No Joule self-heating effects are expected in our test structures at the small bias voltages, about ~1 V, and low current range, about ~1 μA. One can readily estimate the temperature rise, $\Delta T$, from an analytical formula derived from Fourier's law: $\Delta T = P \times R_T = I \times V \times (T_{ox}/K \times L \times W)$. Here, $P$ is the dissipated power in the nanowire, $R_T$ is the thermal resistance of the SiO$_2$ layer, $T_{ox}$ is the thickness of the SiO$_2$ layer, $K_{ox}$ is the thermal conductivity of SiO$_2$, $L$ is the length of the channel, and $W$ is the width of the channel. The formula assumes that the bulk Si substrate is the thermal sink, and the

$SiO_2$ layer with its low thermal conductivity is the thermal barrier. Using the typical values, i.e., $I = 1\,\mu A$, $V = 1\,V$, $T_{ox} = 300\,nm$, $K_{ox} = 1\,W/mK$, $L = 5\,\mu m$, and $W = 300\,nm$, we obtain for the thermal resistance $R_T \sim 0.2$ MK/W and, correspondingly, for the temperature rise $\Delta T \sim 0.2\,K$. We can elaborate the estimate and consider the thermal boundary resistance, $R_C$, between the nanowire and $SiO_2$ layer. It is well known that it is primarily defined by the interface quality rather than specific channel material. Assuming the worst-case scenario of low interface conductance, $G_T = 15\,MW/m^2 K$[44,45], we obtained an extra thermal resistance of ~0.2 MK/W, which doubles the total resistance and temperature rise. On the other side, conduction to metal contacts along the channel may reduce the temperature rise. These simple but reliable estimates show that in our devices, in the considered bias and current ranges, the local heating is negligible. One has to pass currents in the mA range, a factor of ×1000 larger, to induce significant Joule heating. The temperature rise was also assessed using COMSOL software tools for a given device structure and bias voltages. The maximum temperature increase at the hot spot (center of the channel) is approximately 1 K at a voltage drop of 0.9 V. The simulation results confirmed negligible self-heating effects (see Supplemental Materials).

For an independent experimental proof of the absence of local heating effects, we took Raman spectra in situ of the representative devices under the same bias as used in the noise experiments. No characteristic phonon peak shifts, expected for local heating, have been observed (see experimental data in the Supplemental Information). We have previously detected local heating effects with Raman spectroscopy in a range of materials and different device structures, and are confident in the accuracy of such a method[46,47].

## The electronic noise of the CDW condensate in NbS$_3$-II nanowires

We now address another fundamental question: Is the ultra-low noise of sliding condensate unique to the IC-CDW phase of $(TaSe_4)_2I$, or is it a property of many quasi−1D CDW conductors? The early studies of the "broadband" noise in bulk CDW crystals focused on the threshold noise[15–17] and never considered the possibility of noise reduction below the normal electron limit. A few recent studies for $TaS_3$ and $NbSe_3$ found that beyond $V_t$, the noise returns to the normal electron limit[37,48], in contrast to our observation of continuous noise reduction in the sliding regime of $(TaSe_4)_2I$ nanowire devices. Other potential candidates for ultra-low noise transport should include CDW materials where the collective, strongly-correlated current of the sliding electron-lattice condensate is dominant. We experimented with nanowires of NbS$_3$-II, which exists in the CDW phase at RT[32]. The NbS$_3$-II nanowire device fabrication and measurements followed the same protocols as with $(TaSe_4)_2I$ nanowire devices. Figure 6a−f shows transport and noise characteristics for a representative NbS$_3$-II nanowire device. The most important observation is that the normalized spectral density in the NbS$_3$-II device reduces below the normal electron limit, similar to that in the $(TaSe_4)_2I$ devices. The noise spectral density scaling with current is also consistent, i.e., noise reduces with increasing current. It reveals a similar, $S_I / I^2 \propto 1/I$, dependence for the current above 10 μA. We note that the noise reduction in NbS$_3$-II is achieved at temperatures above RT. The latter is important for practical applications, i.e., in interconnects. There is one difference in the noise behavior in the two materials: it appears that the noise spectral density in NbS$_3$-II nanowires shows saturation behavior, even though the noise level is below the normal electron limit. This suggests that we may be reaching the intrinsic electron-lattice condensate noise in NbS$_3$-II nanowires but not in $(TaSe_4)_2I$ nanowire devices (see additional data in Supplementary Materials).

The origin of the ultra-low-noise collective current in $(TaSe_4)_2I$ nanowire, and the lack of noise saturation, may be related to the fact that the CDW transport in this fully-gapped material is described by the

under-damped oscillator model[49,50]. Compared to other CDW materials, $(TaSe_4)_2I$ has the largest energy gap and heaviest CDW effective mass, $m^*/m \approx 10^4$, yielding the lowest CDW damping. The nonlinear CDW conductivity in this material continues to increase rather than saturate due to damping, in the examined bias voltages. The current in NbS$_3$-II shows signs of saturation at higher bias voltages (see Fig. 6b), indicating the onset of stronger CDW current damping, which likely translates to higher intrinsic CDW noise and eventual flattening of the total noise level (see Fig. 6e, f). Based on the recent realization that $(TaSe_4)_2I$ is a Weyl semimetal[30,31], one may also argue that topological scattering suppression has relevance to the noiseless current of the sliding condensate in this material. Surveying any other possible material systems where normalized noise can be suppressed, we noted systems with weak localization effects[51].

It is interesting to compare the noise dependence on bias in $(TaSe_4)_2I$ and NbS$_3$-II nanowires with that in $NbSe_3$ nanowires[37]. The noise in $(TaSe_4)_2I$ and NbS$_3$-II nanowires decreases below the normal electron limit when the CDW sliding current component starts to dominate. The exact nature of multiple polymorphs of $NbS_3$ and the origin of several CDW transitions in NbS$_3$-II is still under investigation[32,33,52,53]. However, both $(TaSe_4)_2I$ and NbS$_3$-II are known to have a large contribution of electron-lattice condensate to the total current after depinning. The electrical conduction is different in $NbSe_3$ nanowires. This partially gapped CDW material always remains metallic, with the significant normal electron contribution to conduction, even at low temperatures[54]. After CDW depinning, the current in $NbSe_3$ nanowires is dominated by the normal electron contribution. Correspondingly, the noise level does not decrease below the normal electron limit[37]. These considerations add to the importance of the present finding of noise reduction and unusual scaling with the current in nanowires made of CDW materials such as $(TaSe_4)_2I$ and NbS$_3$-II.

## Benchmarking with conventional electronics

Finally, to benchmark the noise performance of CDW nanowires, we compared their noise level with that of advanced Si metal-oxide-semiconductor field-effect transistors (MOSFETs), calculated using the standard McWhorter model for a transistor channel with the same area as the nanowires[55–57]. At the conductivity $G = 10^{-5}$ (S), midway through our measurement range, the noise in a $(TaSe_4)_2I$ nanowire in the CDW sliding regime is $S_I / I^2 \sim 1.7 \times 10^{-8}$ (Hz$^{-1}$) at a reference $f = 10$ Hz. This value is lower than the noise in MOSFETs, $S_I / I^2 \sim 3.2 \times 10^{-8}$ (Hz$^{-1}$), with the realistic trap density of $N_t = 10^{20}$ cm$^{-3}$ eV$^{-1}$. Given that our nanowires and devices, fabricated at university facilities, are less perfect compared to commercial MOSFETs, this comparison underscores the promise of CDW-based devices for ultra-low-noise applications.

## Conclusions

In summary, we found that in CDW nanowire conductors made of $(TaSe_4)_2I$ and NbS$_3$-II, the low-frequency noise can drop below the noise limit of normal electronic conductors. The sliding electron-lattice condensate of strongly correlated charge carriers is less noisy compared to the noise of the normal electrons. We observed no residual minimum noise level in our experiments with $(TaSe_4)_2I$, suggesting the ultra-low intrinsic noise of the CDW electron-lattice condensate. The total normalized noise spectral density in the CDW sliding regimes scales inversely with current, a striking difference from conventional metals and semiconductors, where the normalized noise level remains constant with the current. In NbS$_3$-II nanowires, although the noise generated by the CDW condensate saturates the overall noise at higher biases, the CDW noise remains lower than the noise of individual electrons. At higher current densities, the lower intrinsic noise levels of the sliding condensate can benefit CDW conductors by providing better signal-to-noise ratios. The obtained experimental results and insights into the fundamental physics of CDW nanowires could be

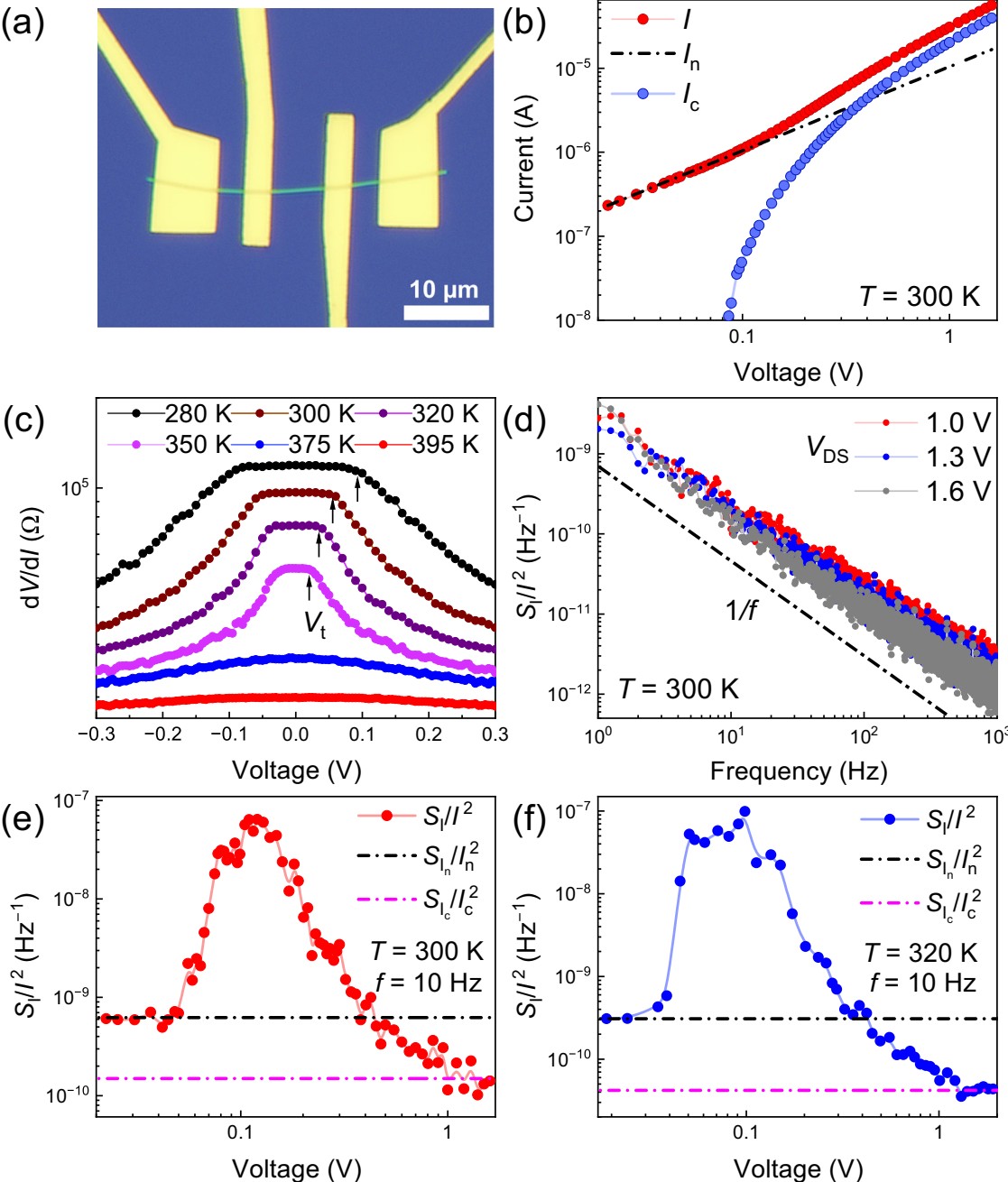

**Fig. 6 | Transport and noise data for NbS$_3$-II nanowire devices. a** Optical microscopy image of an NbS$_3$ nanowire device with several top metal electrodes. The white/dark colors correspond to high/low light reflection. **b** The I−V characteristics of the middle channel in the incommensurate CDW phase at room temperature. The collective current of the CDW sliding condensate is shown with blue circles. **c** Differential resistance dependence on applied bias, between $T = 280$ K and 395 K. The threshold voltage, $V_t$, for depinning increases with decreasing temperature, as expected. **d** The noise spectra at higher biases, when the current is dominated by the CDW condensate. **e** The noise, $S_I/I^2$, at a fixed frequency $f = 10$ Hz $vs$. bias voltage, measured at $T = 300$ K. **f** The same as (**e**), measured at $T = 320$ K. The black and magenta colored straight lines indicate the linear and CDW sliding noise levels, respectively. The total normalized noise in NbS$_3$-II decreases below the normal electron noise limit, like in (TaSe$_4$)$_2$I nanowires, but then shows saturation behavior at high bias, indicating the intrinsic CDW noise level in this material.

transformative for quantum technologies and suggest a way for future noiseless electronics.

## Methods

### Chemical vapor transport synthesis and growth of (TaSe$_4$)$_2$I

On the benchtop, 0.6468 g (3.575 mmol) of Ta powder (Strem, 99.98%) and 1.2511 g (15.85 mmol) of Se powder (Strem, 99.99%) were gently mixed. I$_2$ crystals (0.3270 g, 2.577 mmol, JT Baker, 99.9%) were placed at the bottom of a pre-cleaned and dried fused quartz ampule (~18 × 1 cm length, 10 mm inner diameter, 14 mm outer diameter, ~13 cm³ volume). The mixed Ta and Se powders were then transferred into the ampule, using a glass funnel and anti-static brush to ensure a clean and complete transfer. After loading, the ampule was capped with an adapter and valve, then submerged in an acetonitrile/dry ice bath and subjected to four evacuation/backfilling cycles with Ar(g) using a Schlenk line. The ampule was subsequently sealed under vacuum. The sealed ampule was placed in a horizontal tube furnace, where

the temperature was ramped over 6 h to establish a thermal gradient of 600 °C (source zone) to 550 °C (growth zone). This gradient was maintained for 240 h. Then the ampule was cooled to RT over 12.25 h. Upon completion, 0.5929 g of lustrous, shard-like crystals were recovered from the growth zone and left to sit in a fume hood for 1 h (29.60% isolated yield). These crystals were stored in an Ar-filled glovebox.

### Chemical vapor transport synthesis and growth of $NbS_3$

On the benchtop, 1.4806 g (15.94 mmol) of Nb powder (Strem, 99.99%) and 1.6921 g (52.77 mmol) of S powder (J.T. Baker, ≥99.5%) were gently mixed. This mixture was placed at the bottom of a pre-cleaned and dried fused quartz ampule (~9 cm × 2.2 cm length, 1.9 cm inner diameter, 2.2 cm outer diameter, volume ~33 cm³). A glass funnel and anti-static brush were used to ensure a clean and complete transfer. After loading, the ampule was capped with an adapter and valve, then submerged in an acetonitrile/dry ice bath and subjected to four evacuation/backfilling cycles with Ar(g) using a Schlenk line. The ampule was subsequently sealed under vacuum. The sealed ampule was placed in a horizontal tube furnace, where the temperature was ramped over 3.7 h to establish a thermal gradient of 700 °C (source zone) to 670 °C (growth zone). This gradient was maintained for 240 h. Then the ampule was cooled to RT over 110 h. Upon completion, 3.0247 g of dense, dark silver-gray, wirelike crystals were recovered from throughout the ampule (95.43% isolated yield). These crystals were stored in an Ar-filled glovebox.

### Material characterization of $(TaSe_4)_2I$ samples

$(TaSe_4)_2I$ crystallizes in the tetragonal space group $I422$ with RT lattice parameters of $a = b = 9.5310$ Å, $c = 12.8240$ Å, $V = 1164.932$ Å. Covalently bonded Ta and Se atoms form chiral, helical chains, with I atoms weakly bonded to them along the $c$ axis (Fig. S1a), allowing exfoliation to nanometer-scale thicknesses. SEM imaging was performed using an FEI Teneo FE-SEM at 10 keV with a spot size of 10. Energy-dispersive X-ray spectroscopy (EDS) was performed using an Aztec Oxford Instruments X-MAX$^N$ detector operated at 10 keV with a spot size of 10. For SEM and EDS analysis, the sample was prepared by mounting the as-grown crystals onto a stub using carbon tape and mechanically exfoliating the crystal surfaces with Scotch tape. EDS maps demonstrate homogeneity of the constituent elements (Fig. 1a). In addition, EDS spectra and measured atomic percentages are consistent with stoichiometric $Ta_2Se_8I$ (Fig. S1c). Raman spectroscopy of $(TaSe_4)_2I$ crystals was conducted at RT using a Thermo Fisher DXR Raman Microscope with ×50 magnification and $\lambda = 532$ nm laser excitation at 2 mW. The characteristic peaks of $(TaSe_4)_2I$, being observed at 70, 101, 147, 160, 182, and 271 cm⁻¹, are consistent with previous observations and are sharp, further indicating the high quality of the sample (Fig. S1d). Transmission electron microscopy (TEM) imaging was performed with a JEOL JEM-2100PLUS microscope at 200 kV. Sample preparation: 13.3 mg of $(TaSe_4)_2I$ crystals were probed and bath sonicated in 20 mL ethanol, and the resulting dispersion was drop cast onto a lacey carbon copper-supported grid. High-resolution TEM data align well with the crystal structure of $(TaSe4)_2I$ (note overlay in Fig. S1e) and Fast-Fourier Transforms (FFTs) match theoretical patterns, verifying that the atomic structure of $(TaSe_4)_2I$ is maintained upon exfoliation. Powder X-ray diffraction data were collected using a Bruker D2 Phaser diffractometer equipped with a LYNXEYE XE-T linear position-sensitive detector and Cu Kα ($\lambda = 1.5418$ Å) radiation. Sample preparation: 20.7 mg of $(TaSe_4)_2I$ crystals were finely ground with 11.5 mg of amorphous quartz glass using a mortar/pestle. Rietveld refinement (Fig. S1f) was performed using TOPAS; maintaining the tetragonal $I422$ space group of $(TaSe_4)_2I$, the $a$ parameter was refined to 9.5551 Å and $c$ to 12.7966 Å, for a final cell volume of 1168.331 Å. These results closely match the parameters of ICDD reference pattern 04-011-3118 (cell parameters listed above), with no evidence of additional phases

observed. Goodness of fit was 2.79 due to preferred orientation effects still being present in the ground sample.

### Material characterizations of $NbS_3$-II samples

$NbS_3$-II crystallizes in the monoclinic space group $P2_1/m$ with RT lattice parameters of $a = 9.6509(8)$ Å, $b = 3.3459(2)$ Å, and $c = 19.850(1)$ Å. The structure of $NbS_3$-II is composed of bilayers of sulfur-bridged [$NbS_6$] chains, connected weakly through vdW interactions (Fig. S11a). SEM imaging was performed using an FEI Teneo FE-SEM at 10 keV with a spot size of 10, showing facile cleavage into nanowires (Fig. S11b). EDS was performed using an Aztec Oxford Instruments X-MAX$^N$ detector operated at 10 keV with a spot size of 10. For SEM and EDS analysis, the sample was prepared by mounting the as-grown crystals onto a stub using carbon tape and mechanically exfoliating the crystal surfaces with Scotch tape. EDS maps demonstrate the homogeneity of sulfur and niobium (Fig. S11c), and measured atomic percentages are consistent with a slightly S-deficient sample (Fig. S11d). TEM imaging of exfoliated $NbS_3$ nanowires was conducted with a Thermo Fisher Titan 80–300 probe aberration-corrected scanning transmission electron microscope (STEM) at RT with 200 kV accelerating voltage. This sample was prepared by bath sonication of $NbS_3$ crystals in ethanol, and the resulting dispersion was drop cast onto a lacey carbon copper-supported grid. High-resolution TEM data align well with the crystal structure of $NbS_3$ (note overlay in Fig. S11e) and FFTs match theoretical patterns, verifying that the atomic structure of $NbS_3$ is maintained upon exfoliation. Raman measurement was performed using a Renishaw inVia system in the backscattering geometry with a 633 nm excitation laser.

### Fabrication of the CDW nanowire test structures

The $(TaSe_4)_2I$ and $NbS_3$-II nanowire test structures were prepared by exfoliating high-quality bulk needle-like crystals into thinner nanowire samples of high aspect ratio on top of Si/SiO$_2$ substrates. The nanowire test structures were fabricated in the cleanroom environment with electron beam lithography and electron beam metal deposition. The thickness of the nanowires was on the order of ~50–100 nm, whereas the channel length, $L$, was in the range from 2 µm to 12 µm. The preparation of $(TaSe_4)_2I$ nanowire multi-channel test structures began with cutting small substrates (~0.5 cm × 0.5 cm) from a large SiO$_2$/Si wafer (University Wafer, p-type, Si/SiO$_2$, <100>) and cleaning them multiple times using acetone and isopropyl alcohol (IPA), followed by rinsing with deionized (DI) water. Next, the bulk $(TaSe_4)_2I$ crystal was exfoliated into thin nanowires using the conventional mechanical exfoliation technique on clean substrates. Using an optical microscope, nanowires with varying thicknesses and cross-sectional areas were selected as channels for fabrication. The test structures were fabricated in a class-100 cleanroom. The process began by loading the substrates containing the exfoliated nanowires into a spin coater (Headway Research), spin-coating them twice with a PPMA A4 (Kayaku Advanced Materials, 495 PPMA) solution, and then baking them at 150 °C for 5 min each. Next, the spin-coated samples were placed inside an electron-beam lithography (EBL) system (JEOL JSM 6610), where predesigned patterns for electrodes and contact pads were written at specific accelerating voltages, area doses, and beam currents. Afterward, the samples were developed in a solution (MIBK: IPA, 1:3) for 1 min and rinsed with DI water for 30 s. Immediately after, the samples were transferred to an etcher system (Oxford 80+), where they underwent an Ar-gas plasma-based cleaning process for 30 s to remove any chemical residue from the exposed patterns after the development solution. Next, the samples were placed into an electron-beam evaporation (EBE) system (CHA Mark 40) for metal deposition, where a 10 nm Ti adhesive layer and 90 nm Au metal layers were deposited at rates of 0.5 Å/sec and 1.0 Å/sec, respectively. Finally, the substrates with the test structures and deposited Ti/Au contacts were submerged in acetone for the lift-off process. That completed the

process of fabricating (TaSe$_4$)$_2$I and NbS$_3$-II nanowire test structures with varying nanowire channel lengths defined by the metal electrodes. The (TaSe$_4$)$_2$I device 1, described in most detail in the main text, had a channel width of 350 nm and a thickness of 83 nm.

### Electronic transport and noise measurements in the CDW nanowires

The temperature-dependent transport measurements of the nanowire test structures were conducted inside a cryogenic probe station (Lakeshore TTPX) under high vacuum (-1 × 10$^{-5}$ torr). A semiconductor analyzer (Agilent B1500A) was used for the two-terminal and four-terminal I–V measurements. In the case of the four-terminal measurements of the middle channel, a constant current was applied to the outer two contacts while measuring the voltage drop between the inner two contacts. The contact resistance of the channel was calculated using the difference between the resistances from the two-terminal and four-terminal configurations. For temperature-dependent resistance measurements, the channel current at different temperatures was measured at a constant low applied voltage in the linear region in an automatic measurement set-up using a temperature controller (Lakeshore Model 336). The noise measurements were conducted by placing the device under test (DUT) inside the cryogenic probe station and controlling its ambient temperature. The measurement circuit consisted of a voltage divider circuit with the DUT and a load resistor connected in series with a low-noise DC biasing battery. A potentiometer controlled the voltage drop across the circuit. During the noise measurements, the voltage across the circuit, $V_T$, and the grounded DUT, $V_D$, were measured using digital multimeters (DMM). The current across the DUT, $I_D$, was calculated from the voltage, $V_L = V_T - V_D$, on the load resistor with known resistance, $R_L$. The resistance of the DUT was determined using the voltage reading of the DUT, $V_D$, from the DMM and the calculated current across the load resistor, $I_L = V_L/R_L = I_D$. To measure the noise, the output voltage fluctuation, $\Delta V$, of the circuit was transferred to a voltage preamplifier (SR 560), which amplified the input signal. The amplified voltage signal was then transferred to a dynamic signal analyzer (Photon+), which converted the time-domain signal to the corresponding voltage spectral density, $S_V$, using the FFT. The electricity grid 60 Hz harmonics were removed from the noise spectra. In the noise calculations, the obtained $S_V$ was converted into its equivalent short-circuit current spectral density, $S_I$.

### The theory of $1/f$ noise in CDW systems in the linear approximation

To get an insight into the physics of current fluctuations in CDW conductors, we developed a theory of $1/f$ noise in linear approximation. It includes contributions of three main components of noise and shows the principal possibility of the total noise reduction below the limit of normal electrons. Let us consider the expression for the CDW current based on $J = \rho v$ where $\rho = qn_c$ and $v = \mu_c(E - E_t)$, where $n_c$ is the charge density of the CDW, $\mu_c$ is its mobility, $E$ is the electric field, and $E_t$ is the threshold field. We note that $E = V_{DS}/L$ and $E_t = V_t/L$ where $L$ is the channel length and $V_t$ is the threshold voltage for $V_{DS}$ to cause sliding. Therefore, for a CDW conductor, we obtain:

$$I_D = \frac{Wd}{L} q n_c \mu_c (V_{DS} - V_t)(V_{DS} \geq V_t).$$ (M1)

Here $W$ is the channel width and $d$ is the channel depth. Note that this is a modified Ohm's law for a conductor with a threshold voltage, since the conductivity is $\sigma_C = qn_c\mu_c$.

Starting with Eq. (M1), and adding the normal current, we have:

$$I = G_n V + G_c (V - V_t)\theta(V - V_t),$$ (M2)

where $G_n$ is the normal current conductivity and $G_c = \frac{Wd}{L} qn_c \mu_c$ is the CDW conductivity. Here, $\theta(V - V_t)$ is the unit step function that

ensures the CDW contribution begins only after depinning. We can write the current fluctuation as:

$$\delta I = \delta G_n V + [\delta G_c (V - V_t) - G_c \delta V_t]\theta(V - V_t)$$
$$= \frac{\delta G_n}{G_n}(G_n V) + \left[\frac{\delta G_c}{G_c} G_c (V - V_t) - \frac{\delta V_t}{V_t} G_c V_t\right]\theta(V - V_t)$$ (M3)

Dividing by the total current, we obtain:

$$\frac{\delta I}{I} = \frac{\delta G_n}{G_n} \frac{G_n V}{G_n V + G_c (V - V_t)}$$
$$+ \frac{\delta G_c}{G_c} \frac{G_c (V - V_t)\theta(V - V_t)}{G_n V + G_c (V - V_t)} - \frac{\delta V_t}{V_t} \frac{G_c V_t \theta(V - V_t)}{G_n V + G_c (V - V_t)}.$$ (M4)

As the next step, we square, take the ensemble average, and assume no cross correlation to get to the final expression for noise in the linear model:

$$\frac{\langle \delta I^2 \rangle}{I^2} = \frac{\langle \delta G_n^2 \rangle}{G_n^2} \frac{G_n^2 V^2}{(G_n V + G_c (V - V_t))^2}$$
$$+ \left\{\frac{\langle \delta G_c^2 \rangle}{G_c^2} \frac{G_c^2 (V - V_t)^2}{(G_n V + G_c (V - V_t))^2} + \frac{\langle \delta V_t^2 \rangle}{V_t^2} \frac{G_c^2 V_t^2}{(G_n V + G_c (V - V_t))^2}\right\}\theta(V - V_t).$$ (M5)

$$\frac{\langle \delta I^2 \rangle}{I^2} = \frac{\langle \delta G_n^2 \rangle}{G_n^2} \frac{I_n^2}{I^2} + \left\{\frac{\langle \delta G_c^2 \rangle}{G_c^2} \frac{I_c^2}{I^2} + \frac{\langle \delta V_t^2 \rangle}{V_t^2} \frac{V_t^2}{(V - V_t)^2} \frac{I_c^2}{I^2}\right\}\theta(V - V_t).$$ (M6)

At $V = V_t^-$,

$$\frac{\langle \delta I^2 \rangle}{I^2} = \frac{\langle \delta G_n^2 \rangle}{G_n^2} \quad (V = V_t^-)$$ (M7)

At $V = V_t^+$, the noise jumps up by $\frac{\langle \delta V_t^2 \rangle}{V_t^2} \frac{G_c^2}{G_n^2}$:

$$\frac{\langle \delta I^2 \rangle}{I^2} = \frac{\langle \delta G_n^2 \rangle}{G_n^2} + \frac{\langle \delta V_t^2 \rangle}{V_t^2} \frac{G_c^2}{G_n^2} \quad (V = V_t^+)$$ (M8)

In the limit of large $V \gg V_t$,

$$\frac{\langle \delta I^2 \rangle}{I^2} = \frac{\langle \delta G_n^2 \rangle}{G_n^2} \frac{G_n^2}{(G_n + G_c)^2} + \frac{\langle \delta G_c^2 \rangle}{G_c^2} \frac{G_c^2}{(G_n + G_c)^2}$$
$$+ \frac{\langle \delta V_t^2 \rangle}{V_t^2} \frac{G_c^2}{(G_n + G_c)^2} \frac{V_t^2}{V^2} (V \gg V_t)$$ (M9)

The term proportional to $\frac{\langle \delta V_t^2 \rangle}{V_t^2}$ falls off as $\frac{1}{V^2} \sim \frac{1}{I^2}$. Further, if the fluctuations due to $\delta G_c$ are neglected, then for $V \gg V_t$,

$$\frac{\langle \delta I^2 \rangle}{I^2} = \frac{\langle \delta G_n^2 \rangle}{G_n^2} \frac{G_n^2}{(G_n + G_c)^2} + \frac{\langle \delta V_t^2 \rangle}{V_t^2} \frac{G_c^2}{(G_n + G_c)^2} \frac{V_t^2}{V^2} (V \gg V_t)$$ (M10)

so that we reach a limit set by the normal current noise in parallel with a noiseless channel, i.e., for $V \to \infty$,

$$\frac{\langle \delta I^2 \rangle}{I^2} = \frac{\langle \delta G_n^2 \rangle}{G_n^2} \frac{G_n^2}{(G_n + G_c)^2} (V \to \infty).$$ (M11)

Compared to Eq. (1) of the main text, the derived phenomenological theory of $1/f$ noise of CDW conductors has a third term, related to the fluctuations in the threshold voltage. This is an extrinsic noise component since the threshold is related to the defects pinning the CDW condensate. The linear theory does not give an accurate

description of the observed noise, since the conductance is nonlinear, but provides an insight into the physics of current fluctuations.

## Data availability
The data that support the findings of this study are available from the corresponding author upon request.

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

## Acknowledgements

The work at UCLA was supported by the Vannevar Bush Faculty Fellowship (VBFF) to A.A.B. under the Office of Naval Research (ONR) contract N00014-21-1-2947 on One-Dimensional Quantum Materials. The work at UCR and the University of Georgia was supported, in part, via the subcontracts of the ONR project N00014-21-1-2947. HRTEM was performed using the JEOL 2100PLUS microscope, acquired with funding from the National Institutes of Health through grant 1S10OD034282-01. S.R. acknowledges partial support by the European Research Council (ERC) Project No. 101053716. The authors acknowledge useful discussions with M. Taheri and J. Brown at UCLA. The nanofabrication of the test structures was performed in the California NanoSystems Institute (CNSI).

## Author contributions

A.A.B. conceived the idea, coordinated the project, contributed to the model development and data analysis, and led the manuscript preparations. S.G. fabricated the nanowire test structures, conducted electronic transport and noise measurements, and contributed to model development and data analysis. R.L. led the nonlinear model development and analysis. S.R. contributed to the model development and data analysis. Z.E.N. conducted in situ Raman measurements and COMSOL simulations. N.S. synthesized bulk crystals using the CVT method and performed microscopy and materials characterization. T.S. supervised material synthesis and contributed to materials characterization. All authors participated in the manuscript preparation.

## Competing interests

The authors declare no competing interests.
