## [Transparent Peer Review File · Nature Communications]

A Quieter State of Charge and Ultra-Low-Noise of the Collective Current in Quasi-1D Charge-Density-Wave Nanowires

Corresponding Author: Professor Alexander Balandin

Version 0:

Reviewer comments:

Reviewer #1

(Remarks to the Author)

The manuscript by Ghosh et al., presents a detailed investigation of noise behavior in a nanoscale charge density wave system. The main claim of the manuscript is that, in the sliding mode of CDW, the normalized noise drops below that of normal electrons. The study and the results are of interest to researchers working in CDW materials in general and to researchers exploring low-noise electronic systems.

Overall, the manuscript presents high quality data, a systematic study and the results are interesting. However, I have a couple of concerns and suggestions for improvements:

(1) Well above V_t , say $V \sim 1.0V$, it is possible that Joule heating plays an important role given the dimensions of the nanowires and the current generated. It will be good to estimate the effect of this and include a discussion. This could be relevant since the contribution from normal electrons (Figure 1 e, for example) is assumed to be unchanged at all voltages.

(2) Figure 2 needs improvement: a Lorentzian bulge is shown in 2(b) in the depinning region whereas 2(a) shows only a smaller frequency range below where the bulge was observed in (b). It would be important to show data from comparable ranges in (a) and (b).

(3) It is not clear if the Lorentzian bulge persists at other voltages than shown in 2(b). Additional data at various voltages in 2(b) can be helpful as well as a discussion from the authors on the evolution of the bulge with voltage.

(4) Four different bias regions (Figure 2) were mentioned with handwaving assignments based on the behavior. It will be helpful to mark the regions in the figures and to state the voltage ranges to avoid confusion.

(5) It will be helpful to state clearly how the normal and collective components of I and G are calculated, the explanation surrounding Figure 3 is confusing and could be improved.

(6) A major concern is about the author's earlier work on noise behavior in another CDW system (reference 36). The observation of the noise magnitude trending significantly lower in the sliding CDW regime has been reported in 36. It will be important for the authors to clearly provide the additional understanding and/or newer result in the present manuscript.

Reviewer #2

(Remarks to the Author)

As known for decades, for some quasi -1D materials with a charge density wave (CDW) ground state, above a threshold value, the pinned CDW can move and contribute to a collective current - static and modulated at a given frequency associated to $1/f$ low frequency noise (unfortunately called narrow and broad band noise).

The manuscript reports studies of non-linear transport properties, specifically noise, in $(\text{TaSe}_4)_2\text{I}$ nanostructures, the Peierls transition of which occurs at $T_P = 245\text{K}$. Data are presented for two different devices at a single temperature below T_P . The authors show that far away from the threshold (10 times its value at the given temperature), the noise is suppressed below the value of normal charge carriers, and that the noise of the CDW current is negligible. Less elaborated a similar conclusion was suggested (see Ref.36) in noise measurements in NbSe_3 which at the difference of $(\text{TaSe}_4)_2\text{I}$ remains metallic below the Peierls transitions.

These conclusions are questionable as discussed in the following:

-in Fig.1d the variation of the differential resistance in a semi-log plot is shown versus the applied voltage at different temperatures: clearly the resistance at fixed T is not constant indicating a heating effect. At $T=120\text{K}$ the drop of resistance at the threshold (see the arrow) is even not visible. No estimation of thermodynamics appears in the paper.

- While data in Fig.1d concern the small values of V ($V < 0,2\text{V}$) those in Fig.1e are extended to 1V . How are the dV/dI characteristics till 1V in order to see possible jumps or inhomogeneities masked in the log-log I - V curves in fig.1e ?.

-Data in Fig1e show that the CDW current (evaluated from the extended (dashed line) Ohmic law (without taking care of any thermal effect) follows a power law without saturation till more than 10 times the threshold value. How to reconcile this result with theories which evaluate the high field limit of the CDW conductance?.

-Eq. 2 is the combination of Eq.(5.6) and 5.14) from Ref.1

The first term results from the "single particle" model – classical particle moving in a sinusoidal washboard potential -in the case where the voltage is regulated yields $j \propto (E-E_t)^{1/2}$, (however in the part but" Electronic transport and noise measurement in the $(\text{TaSe}_4)_2\text{I}$ nanowires" it seems that measurements were performed with a regulated current. In that case the threshold is manifested by an infinite negative differential resistance that is not the case in Fig.1d).

The second term provides from a different model due to dynamical critical phenomenon with the CDW velocity or the CDW current is a power law $(E-E_t)$ with an exponent $3/2$ to be compared with $1/2$ in the single particle model (however see and 2 in eq.2). No real justification is made for the addition of two different threshold current with non-linear voltages with different exponents, even a third one - see figure caption of Extended data Fig.6).

From their Eq.2 the authors derive the noise spectral density formed of 3 terms: noise of the normal current, noise of the fluctuations of the threshold voltage (near V_t) and fluctuations of the CDW conductance. They found that the noise spectra can be accurately fitted with only the noise amplitude of the normal current and that of threshold fluctuations but consequently without noise for the CDW conductance. However the values of different parameters such these three values of V_t are not indicated.

I have taken consideration of this work but I consider that the conclusion of a CDW condensate moving without noise is not justified based on the presented data. There are too many uncertainties either in the experimental measurements and in the theoretical and fitting process which raise some, in my sense, reasonable doubts on the conclusions.

Thus I will not recommend publication of this work in Nature Communications

Reviewer #3

(Remarks to the Author)

Reviewer #4

(Remarks to the Author)

The manuscript **NCOMMS-25-46228-T** by Subhajit Ghosh et al. describes a study of low-frequency electronic noise suppression in quasi-one-dimensional charge-density-wave (CDW) systems, specifically in $(\text{TaSe}_4)_2\text{I}$ nanowires, under conditions of sliding Frohlich condensate current.

The reported suppression of low-frequency $1/f$ noise in the sliding CDW regime is highly unusual and underlines a significant difference from conventional transport behavior in resistive materials. In this respect, the linear decrease of normalized noise spectral density, as opposed to the typical constant value in normal conductors, suggests a fundamentally different noise mechanism linked to collective transport. This is an indicator of new physics in the correlated transport regime. Moreover, the connection to real-world technological relevance, emphasizing technological relevance to quantum devices, low-noise communication, and sensing, strengthens the case for broad impact.

In the context of existing literature, the submitted work appears scientifically novel in its experimental observation of sub-thermal electronic noise in the sliding CDW regime, a behavior not established in previous CDW literature where broadband

1/f noise has been usually interpreted as a consequence of CDW dynamics near depinning. Moreover, recent studies, exploring CDW systems in reduced dimensions (nanowires, thin films), are concerned with transport phenomena but not noise suppression, as the case of the article here considered.

Although the work appears sufficiently novel and innovative for publication in Nature Communications, two important points deserve more clarification and should be strongly taken into account before a resubmission.

(I) The first point concerns the fact that the notion of using strong correlations to suppress noise, especially leveraging a collective mode like Frohlich conduction, is still underexplored and seems to be largely speculative. The authors need to better clarify this aspect, demonstrating that the theory developed is based on solid principles and the results obtained are not just material-specific observations. In this respect, further experimental investigations showing an analogous noise behavior as the one here reported, performed on different systems undergoing a similar CDW regime, could be very useful to address such a criticism.

(II) The second point, which is somehow connected to the first one, regards an improvement of the experimental robustness and generality of the work, in order to elevate its impact from a specialized CDW study to a wider platform as, for example, for quantum/nanoelectronics.

As minor comment, due to the high visibility and high impact of Nature Communications journal, my suggestion to the authors is to mention other systems and materials where an evident reduction of the 1/f noise level has been observed. This is the case, for example, of low-dimensional systems and topological insulators. A detailed description of key similarities and key differences in noise suppression mechanisms could also be given.

For all the reasons specified above, in my opinion, the paper is not suitable for publication in Nature Communications in its present form. A major revision, by addressing all the points indicated, is needed before the manuscript can be reconsidered.

Version 1:

Reviewer comments:

Reviewer #1

(Remarks to the Author)

The comments from my earlier review have been satisfactorily addressed. The revised manuscript and supplement information are vastly improved from the previous version.

My only comment is for the authors to consider adding their comments about the novelty of this work compared to previous publication (NbSe₃). They have provided the arguments in the response which I find to be satisfactory, but it will be helpful to have them prominently stated in the revised manuscript.

With that, I recommend publication.

Reviewer #2

(Remarks to the Author)

I acknowledge the authors for the improvements they made in the revised version of their paper including in particular data on the CDW compound NbS₃ with its Peierls transition temperature above room temperature and their phenomenological theory of 1/f noise in Supplementary.

However I did not get convincing answers to the questions raised in my first report, specifically concerning heating effects. The fact that no softening of the phonon modes in Raman experiments is, for the authors, the direct proof of the absence of Joule heating. But the more simple and direct proof is the electric field dependence of the normal resistance which in the pinned CDW state below the threshold should keep its normal value. The activated temperature dependence of the resistance in the normal state of these fully gapped CDW materials is a very sensitive probe of possible heating effects.

No explanation has been given on the electric field dependence of the differential resistance below threshold as shown in Fig1d in a semilog plot in the original manuscript. Opportunely the updated rescaled Fig1d in the revised manuscript has in some way erased this dependency. However this effect seems to be less important in NbS₃ as seen in Fig6c (but data in a linear plot would be important).

This question is of a huge importance since the way to extract the CDW noise results from the subtraction from the total noise of independent noise from the normal current and noise from fluctuations in the threshold voltage. Any change in the

normal current noise may affect the result.

It appears difficult to ascertain as the authors state that Joule heating does not play any role.

I keep the conclusions of my first report

Reviewer #3

(Remarks to the Author)

Reviewer #4

(Remarks to the Author)

The revised manuscript NCOMMS-25-46228A by Subhajit Ghosh et al. has been improved, with respect to the original submission, and my criticisms have been adequately addressed. In particular, additional data of measurements performed on a different system (NbS₃ nanowires) undergoing a similar CDW regime than (TaSe₄)₂I nanowires have been included in this revised version, showing that the results obtained are not just material-specific observations but are more general.

However, I do not agree with one statement of the authors (end of Page 8 of the revised version) where they assert that: "*in this work, it is reported for the first time the observation of a decreasing noise level by increasing the current density*". While, I remember other works in literature reporting systems where a decrease of the noise level is observed with an increasing bias current. This is the case of disordered LBMO thin films or the case of systems undergoing weak-localization effects (<https://doi.org/10.3390/coatings11010096>). My suggestion to the authors is to take into account also these experimental results already published.

In the same review paper (<https://doi.org/10.3390/coatings11010096>), moreover, there is also a section reporting a detailed study of fluctuation mechanisms induced by charge density waves conduction in PCMO epitaxial thin films. The authors should consider these investigations by evidencing similarities or differences between their conclusions and the other ones already published.

Version 3:

Reviewer comments:

Reviewer #2

(Remarks to the Author)

The main text of this last revised version has been slightly amended with additions in Supplementary. To get a final point to this tangled question of Fig.1d, it is now recognized that the data in this figure do not concern Device 1 and are just an illustration from depinning obtained from a different sample (that has to be indicated in the figure caption). Note however that the new fig1d exhibits a T dependance (non-uniform) of the threshold field different of that in the former fig1d in the original and first revised version.

Heating effects may affect measurements for any bias voltage: above threshold they are hidden and included in the non-linear variation of the resistance; below threshold they are detected from the non constant differential resistance. These effects are always present even if their consequences can have a limited effect.

Even if I feel uncomfortable with some assumptions/conclusions in this paper, I acknowledge the work done and I would not oppose the publication of this manuscript. Nevertheless, if this work meets some interest in the community, it can be refuted in future publications

I list below some points that can be considered by the authors; it would be worth to integrate their answers in a final text:

- All the analysis is based on the assumption that the fluctuations of the normal and collective fluctuations are independent without cross-correlation. The green line in fig.4c indicates the constant value of noise of electrons as a function of bias voltage. It might be useful to ascertain this hypothesis and beyond the two-fluid model to take into account the CDW-normal carrier interactions

- Eq.2 gathers two different models: one described as a single particle model which implies a kink or a negative differential resistance at E_t and the other one treated as a dynamical critical phenomena. It would be necessary to be more explicit on the theoretical basis which leads to Eq. 2, beyond reference 40. To what extent of E the $(E - E_t)$ is valid?. How this equation is justified?. Even if the fits with the experimental data look of good quality, how to understand that the exponent, α , determined by the fits is around 1 away from the $\frac{1}{2}$ value for the single particle model and $\frac{3}{2}$ for the critical phenomenon.

- With the maximum voltage used, only the noise due to fluctuations in V_t is observed. That lead the authors to consider this picture as the behavior of ideal CDW sliding as a dissipation-less, noise free process. What about the ac field generated in the sliding?.

- Concerning $(\text{TaSe}_4)_2\text{I}$ and NbS_3 the aim was to compared fully-gapped CDW materials. NbS_3 in the phase II has 3 CDW transitions at $\text{TP}_0 = 460\text{K}$, $\text{TP}_1 = 360\text{K}$ and $\text{TP}_2 = 155\text{K}$. If the conductivity above TP_0 is of hopping conductivity type, the conductivity below TP_1 is very weakly T dependent - see Fig.8 in PRB95, 035611. There are 8 chains in the unit cell and even if the structure is complicated it appears that the electron condensation appears successively on distinct chains (similarly to NbSe_3 and monoclinic and orthorhombic TaS_3). NbS_3 below TP_1 is not fully gapped CDW material . The fully dielectric state is below TP_2 . The best compound with an unique CDW and a fully-gapped for comparison with $(\text{TaSe}_4)_2\text{I}$ would be blue bronze ($\text{K}_0.3\text{MoO}_3$).

RESPONSE TO THE REVIEWERS' COMMENTS

REVIEWER COMMENTS

Reviewer #1 (Remarks to the Author):

The manuscript by Ghosh et al., presents a detailed investigation of noise behavior in a nanoscale charge density wave system. The main claim of the manuscript is that, in the sliding mode of CDW, the normalized noise drops below that of normal electrons. The study and the results are of interest to researchers working in CDW materials in general and to researchers exploring low-noise electronic systems.

Overall, the manuscript presents high quality data, a systematic study and the results are interesting.

RESPONSE: We thank the Reviewer for stating that our manuscript “presents high-quality data, a systematic study, and the results are interesting.”

However, I have a couple of concerns and suggestions for improvements:

(1) Well above V_t , say $V \sim 1.0V$, it is possible that Joule heating plays an important role given the dimensions of the nanowires and the current generated. It will be good to estimate the effect of this and include a discussion. This could be relevant since the contribution from normal electrons (Figure 1 e, for example) is assumed to be unchanged at all voltages.

RESPONSE: The reviewer correctly notes that any **significant** Joule heating during transport measurements would affect the results. However, we can state with high confidence that **Joule heating does not play any role in our measurements**. We

used bias voltages below or around 1 V with the current levels always remaining in the micro-Ampere range, resulting in minimal power dissipation.

In the revised manuscript, we included analytical estimates of the thermal resistance of the structure and the resulting temperature rise, which was below 1 K at 0.7 V bias, when the noise drops below the normal electron level. We also conducted COMSOL modeling to verify this estimate. We included noise data for a new material system, NbS₃, at room temperature. Additionally, in situ bias-dependent Raman measurements were conducted on both (TaSe₄)₂I and NbS₃ nanowire devices, where we monitored phonon peak positions *in situ*. The local heating would reveal itself *via* softening of phonon peaks. None was observed at the highest bias voltages, presenting a direct proof of the absence of Joule heating. We discussed the temperature rise estimation and the in-situ Raman behavior in the revised text and included the relevant figures in the Supplemental file. In the revised manuscript, we added the following section.

“In the data analysis, it is also important to exclude local heating effects. No Joule self-heating effects are expected in our test structures at the small bias voltages, about ~1 V, and low current range, about ~ 1 μ A. One can readily estimate the temperature rise, ΔT , from an analytical formula derived from Fourier’s law: $\Delta T = P \times R_T = I \times V \times (T_{ox}/K \times L \times W)$. Here, P is the dissipated power in the nanowire, R_T is the thermal resistance of the SiO₂ layer, T_{ox} is the thickness of the SiO₂ layer, K_{ox} is the thermal conductivity of SiO₂, L is the length of the channel, and W is the width of the channel. The formula assumes that the bulk Si substrate is the thermal sink, and the SiO₂ layer with its low thermal conductivity is the thermal barrier. Using the typical values, *i.e.*, $I=1 \mu$ A, $V=1$ V, $T_{ox} = 300$ nm, $K_{ox} = 1$ W/mK, $L= 5 \mu$ m, and $W=300$ nm, we obtain for the thermal resistance $R_T \sim 0.2$ MK/W and, correspondingly, for the temperature rise $\Delta T \sim 0.2$ K. We can elaborate the estimate and consider the thermal boundary resistance, R_C , between the nanowire and SiO₂ layer. It is well known that it is primarily defined by the interface quality rather than specific channel material. Assuming the worst-case scenario of low interface conductance, $G_T = 15$ MW/m²K^{44,45}, we obtained an extra thermal resistance of ~ 0.2 MK/W, which doubles the total resistance and temperature rise. On the other side, conduction to metal contacts along the channel

may reduce the temperature rise. These simple but reliable estimates show that in our devices, in the considered bias and current ranges, the local heating is negligible. One has to pass currents in the mA range, a factor of $\times 1000$ larger, to induce significant Joule heating. The temperature rise was also assessed using COMSOL software tools for a given device structure and bias voltages. The maximum temperature increase at the hot spot (center of the channel) is approximately 1 K at a voltage drop of 0.9 V. The simulation results confirmed negligible self-heating effects (see Supplemental Materials).

For an independent experimental proof of the absence of local heating effects, we took Raman spectra *in situ* of the representative devices under the same bias as used in the noise experiments. No characteristic phonon peak shifts, expected for local heating, have been observed (see experimental data in the Supplemental Information). We have previously detected local heating effects with Raman spectroscopy in a range of materials and different device structures, and are confident in the accuracy of such a method^{46,47}.”

Regarding the second part of the question, we note that the constant noise level of the normal electrons is not an assumption. It is an established experimental fact (in our measurements, and well before us), which comes from the linear response theory and relates to the fact that current does not drive the fluctuations but only makes them visible via Ohm's law (see page 499 of P. Dutta and P. M. Horn: Low-frequency fluctuations: 1/f noise, Reviews of Modern Physics, Vol. 53, No. 3, July 1981). Mathematically, it means that for normal current $S_I/I^2 = S_V/V^2 = \text{constant}$. This fact also emphasizes the importance of our results and their fundamental nature. We observed that the scaling of 1/f noise of the sliding CDW condensate with the voltage is different. In the revised manuscript, we added an extra explanation to clarify this point.

“The fact that we observed a different dependence on the normalized noise spectral density in a CDW conducting channel is of great importance. It is well-known that the noise level, *i.e.*, normalized noise spectral density, in passive resistors does not depend on current: $S_I/I^2 \propto \text{constant}$ (or $S_I \propto I^2$). This fundamental property originates in the linear response theory³⁸. In terms of physics, it means that the current does not drive the resistance fluctuations in the sample

but rather makes them visible via Ohm's law. The fact that $S_I \propto I^2$ allowed for the introduction of the phenomenological Hooge formula, $S_I/I^2 \sim \alpha_H/fN$, where α_H is the Hooge parameter, and N is the number of charge carriers in the resistor. There are known cases of deviation from $S_I/I^2 \propto \text{constant}$ dependence in specific electronic devices, like diodes³⁹. In this work, we report the first observation of as $S_I/I^2 \propto 1/I$ scaling of noise in a generic resistor with two metal contacts.”

(2) Figure 2 needs improvement: a Lorentzian bulge is shown in 2(b) in the depinning region whereas 2(a) shows only a smaller frequency range below where the bulge was observed in (b). It would be important to show data from comparable ranges in (a) and (b).

RESPONSE: At higher frequencies, the 1/f noise level reaches the background noise floor of the thermal noise or shot noise (at low temperature in some devices). The noise spectrum can be closer to the noise floor when measured at very low biases. Conventionally, the 1/f noise is shown for the relevant frequency range where it is above the thermal floor at different bias ranges (this explains the name “excess noise” for 1/f noise). It is common to report the 1/f noise level at 1 Hz or 10 Hz. In most cases, we reported noise over the 1 kHz range. Since our primary focus is on low-frequency noise behavior, specifically around 10 Hz, which is most relevant to our analysis, we have emphasized that region accordingly. We added more data to the Supplemental Materials regarding the noise spectral behavior of other devices to highlight reproducibility.

(3) It is not clear if the Lorentzian bulge persists at other voltages than shown in 2(b). Additional data at various voltages in 2(b) can be helpful as well as a discussion from the authors on the evolution of the bulge with voltage.

RESPONSE: We have included an additional figure in the revised manuscript to highlight the noise behavior across the depinning transition and added an extra

discussion of the evolution of Lorentzian bulges with bias voltage to the revised manuscript. As shown in the revised Figure 3, the noise spectrum shifts from a purely $1/f$ pattern at lower bias to one showing Lorentzian bulges superimposed on the $1/f$ background at intermediate bias levels, indicating the depinning transition. At higher biases, as the device approaches the sliding regime, the spectrum returns to a $1/f$ noise type, with the Lorentzian features fading and eventually vanishing. This pattern confirms that the origin of the Lorentzian component is related to threshold fluctuations. Additional noise measurements on NbS_3 also confirmed Lorentzian bulges near depinning, which disappear in the sliding regime.

“Figure 3 (a-f) illustrates the evolution of the noise spectra in the bias region II, near the CDW depinning threshold, indicating the association of the Lorentzian features with the depinning. The Lorentzian noise component diminishes as the bias voltage surpasses the depinning threshold and enters the CDW sliding regime. The observed Lorentzian bulges originate from threshold fluctuations; they are associated with the system fluctuations between the pinned and depinned states. In mathematical description, the transitions between two phases in the material are similar to those in the two-level systems, which results in the Lorentzian features in the noise spectra. Additional data is available in the Supplementary Materials.”

(4) Four different bias regions (Figure 2) were mentioned with handwaving assignments based on the behavior. It will be helpful to mark the regions in the figures and to state the voltage ranges to avoid confusion.

RESPONSE: The four regions are introduced in Figure 2 e) based on noise dependence on bias voltage. It is a conventional approach in noise data analysis (see for example, G. Liu, et al., “Low-frequency current fluctuations and sliding of the charge density waves in two-dimensional materials,” *Nano Letters*, 18, 3630 (2018)). The voltage range may change from device to device due to dimensional differences, which affect the depinning. The main feature of $S_V(V)$ remains the same. We marked the regions more clearly in the revised manuscript.

(5) It will be helpful to state clearly how the normal and collective components of I and G are calculated, the explanation surrounding Figure 3 is confusing and could be improved.

RESPONSE: In the revised manuscript, we significantly expanded the description and provided the details in the main text rather than in the Extended Data (as it was in the original submission). Additionally, a comprehensive explanation of the transport parameter extraction and noise modeling procedures has been included in the Supplemental Materials.

(6) A major concern is about the author's earlier work on noise behavior in another CDW system (reference 36). The observation of the noise magnitude trending significantly lower in the sliding CDW regime has been reported in 36. It will be important for the authors to clearly provide the additional understanding and/or newer result in the present manuscript.

RESPONSE: We appreciate that the Reviewer found and read our paper S. Ghosh et al., “The noise of the charge density waves in quasi-1D NbSe₃ nanowires — contributions of electrons and quantum condensate,” Appl. Phys. Rev. 11, 21405 (2024). In that paper, we analyzed the noise behavior in NbSe₃. This is a different CDW material, which is **not fully gapped**. For this reason, the current is always dominated by normal electrons with some contribution from CDWs. In that paper, we focused on noise specifically near the depinning. We have attempted to study the noise at higher bias levels. **However, we have not managed to demonstrate the total noise reduction below the noise level of normal electrons.** Please see Figure 7: the red and gray data points at the far-right merge! We stated this in the text as “**It is important to note that at high currents, the noise of the normal electron current and total noise levels coincide.**” The physical reason for this is that NbSe₃ material is not fully gapped. The current is dominated by the normal electrons. We also did not have a theory to analyze the noise accurately. In extracting the noise of the sliding CDW in that past

work, we simply subtracted the noise at low bias (before CDW depinning) from the total noise. This is a rough approximation. It served as an indication that the CDW noise is reducing and **gave us an idea to study the fully gapped CDW materials and develop a theoretical framework to analyze noise**. However, again, we never reached the total noise level below the normal electrons, the major task. We never observed noise scaling with the current as $S/I^2 \sim 1/I$. These two major tasks required investigation of different material systems, **more than a year of hard work**. **The fundamental discovery of the total noise below the normal electron level and its unusual scaling with the current is reported in the present manuscript for the first time.**

Our new results present a major step forward and novelty: (1) clear demonstration of the noise below normal electron noise (fundamental science discovery); (2) scaling of the CDW condensate noise with bias voltage, which is completely different from scaling of the noise of normal electrons; (3) comprehensive phenomenological theory of $1/f$ noise in CDW materials (something that has not been done before after 50 years of investigations). To add on top, we included data for another quasi-1D material, NbS₃, which shows CDW sliding and noise reduction at room temperature. The additional noise analysis of NbS₃ also suggested that the sliding CDW condensate is inherently less noisy than that of thermalized single electrons. We feel that our manuscript has sufficient novelty for a couple of Nature family publications.

Reviewer #2 (Remarks to the Author):

As known for decades, for some quasi -1D materials with a charge density wave (CDW) ground state, above a threshold value, the pinned CDW can move and contribute to a collective current - static and modulated at a given frequency associated to $1/f$ low frequency noise (unfortunately called narrow and broad band noise).

The manuscript reports studies of non-linear transport properties, specifically noise, in $(\text{TaSe}_4)_2\text{I}$ nanostructures, the Peierls transition of which occurs at $T_P = 245\text{K}$. Data are presented for two different devices at a single temperature below T_P .

RESPONSE: For the original submission, we tested five devices, and the results were consistent. We included two different devices at two different temperatures (180 K and 200 K) in the original manuscript owing to space limitations. In the revised manuscript, we included data for the third $(\text{Ta}_2\text{Se}_4)_2\text{I}$ device and data for a different material system NbS_3 , where we observed a similar effect at room temperature. We included several different temperatures. **We are confident in the reproducibility of the data.**

The authors show that far away from the threshold (10 times its value at the given temperature), the noise is suppressed below the value of normal charge carriers, and that the noise of the CDW current is negligible. Less elaborated a similar conclusion was suggested (see Ref.36) in noise measurements in NbSe_3 which at the difference of $(\text{TaSe}_4)_2\text{I}$ remains metallic below the Peierls transitions.

RESPONSE: Regarding the Ref. 36 (original manuscript numbering), we repeat the response given to Reviewer #1. We appreciate that the Reviewer found and read our paper S. Ghosh et al., "The noise of the charge density waves in quasi-1D NbSe_3 nanowires — contributions of electrons and quantum condensate," Appl. Phys. Rev. 11, 21405 (2024). In that paper, we analyzed the noise behavior in NbSe_3 . This is a different CDW material, which is **not fully gapped**. For this reason, the current is

always dominated by normal electrons with some contribution from CDWs. In that paper, we focused on noise specifically near the depinning. We have attempted to study the noise at higher bias levels. **However, we have not managed to demonstrate the total noise reduction below the noise level of normal electrons.** Please see Figure 7: the red and gray data points at the far-right merge! We stated this in the text as **“It is important to note that at high currents, the noise of the normal electron current and total noise levels coincide.”** The physical reason for this is that NbSe₃ material is not fully gapped. The current is dominated by the normal electrons. We also did not have a theory to analyze the noise accurately. In extracting the noise of the sliding CDW in that past work, we simply subtracted the noise at low bias (before CDW depinning) from the total noise. This is a rough approximation. It served as an indication that the CDW noise is reducing and **gave us an idea to study the fully gapped CDW materials and develop a theoretical framework to analyze noise.** However, again, we never reached the total noise level below the normal electrons, the major task. We never observed noise scaling with the current as $S_I/I^2 \sim 1/I$. These two major tasks required investigation of different material systems, **more than a year of hard work.** **The fundamental discovery of the total noise below the normal electron level and its unusual scaling with the current is reported in the present manuscript for the first time.**

Our new results present a major step forward and novelty: (1) clear demonstration of the noise below normal electron noise (fundamental science discovery); (2) scaling of the CDW condensate noise with bias voltage, which is completely different from scaling of the noise of normal electrons; (3) comprehensive phenomenological theory of $1/f$ noise in CDW materials (something that has not been done before after 50 years of investigations). To add on top, we included data for another quasi-1D material, NbS₃, which shows CDW sliding and noise reduction at room temperature. The additional noise analysis of NbS₃ also suggested that the sliding CDW condensate is inherently less noisy than that of thermalized single electrons. We feel that our manuscript has sufficient novelty for a couple of Nature family publications.

These conclusions are questionable as discussed in the following:

RESPONSE: We hope that our response and additional data provided in the revised manuscript will remove the questions.

-in Fig.1d the variation of the differential resistance in a semi-log plot is shown versus the applied voltage at different temperatures: clearly the resistance at fixed T is not constant indicating a heating effect. At T=120K the drop of resistance at the threshold (see the arrow) is even not visible. No estimation of thermodynamics appears in the paper.

RESPONSE: We would like to state clearly that there are **NO** heating effects in our measurements. Below, we repeat the response given to Reviewer 1.

The reviewer correctly notes that any **significant** Joule heating during transport measurements would affect the results. However, we can state with high confidence that **Joule heating does not play any role in our measurements**. We used bias voltages below or around 1 V with the current levels always remaining in the micro-Ampere range, resulting in minimal power dissipation.

In the revised manuscript, we included analytical estimates of the thermal resistance of the structure and the resulting temperature rise, which was below 1 K at 0.7 V bias, when the noise drops below the normal electron level. We also conducted COMSOL modeling to verify this estimate. We included noise data for a new material system, NbS₃, at room temperature. Additionally, *in situ* bias-dependent Raman measurements were conducted on both (TaSe₄)₂I and NbS₃ nanowire devices, where we monitored phonon peak positions *in situ*. The local heating would reveal itself *via* softening of phonon peaks. None was observed at the highest bias voltages, presenting a direct proof of the absence of Joule heating. We discussed the temperature rise estimation

and the in-situ Raman behavior in the revised text and included the relevant figures in the Supplemental file. In the revised manuscript, we added the following section.

“In the data analysis, it is also important to exclude local heating effects. No Joule self-heating effects are expected in our test structures at the small bias voltages, about ~ 1 V, and low current range, about ~ 1 μ A. One can readily estimate the temperature rise, ΔT , from an analytical formula derived from Fourier’s law: $\Delta T = P \times R_T = I \times V \times (T_{ox}/K \times L \times W)$. Here, P is the dissipated power in the nanowire, R_T is the thermal resistance of the SiO₂ layer, T_{ox} is the thickness of the SiO₂ layer, K_{ox} is the thermal conductivity of SiO₂, L is the length of the channel, and W is the width of the channel. The formula assumes that the bulk Si substrate is the thermal sink, and the SiO₂ layer with its low thermal conductivity is the thermal barrier. Using the typical values, *i.e.*, $I=1$ μ A, $V=1$ V, $T_{ox} = 300$ nm, $K_{ox} = 1$ W/mK, $L= 5$ μ m, and $W=300$ nm, we obtain for the thermal resistance $R_T \sim 0.2$ MK/W and, correspondingly, for the temperature rise $\Delta T \sim 0.2$ K. We can elaborate the estimate and consider the thermal boundary resistance, R_C , between the nanowire and SiO₂ layer. It is well known that it is primarily defined by the interface quality rather than specific channel material. Assuming the worst-case scenario of low interface conductance, $G_T = 15$ MW/m²K^{44,45}, we obtained an extra thermal resistance of ~ 0.2 MK/W, which doubles the total resistance and temperature rise. On the other side, conduction to metal contacts along the channel may reduce the temperature rise. These simple but reliable estimates show that in our devices, in the considered bias and current ranges, the local heating is negligible. One has to pass currents in the mA range, a factor of $\times 1000$ larger, to induce significant Joule heating. The temperature rise was also assessed using COMSOL software tools for a given device structure and bias voltages. The maximum temperature increase at the hot spot (center of the channel) is approximately 1 K at a voltage drop of 0.9 V. The simulation results confirmed negligible self-heating effects (see Supplemental Materials).

For an independent experimental proof of the absence of local heating effects, we took Raman spectra *in situ* of the representative devices under the same bias as used in the noise experiments. No characteristic phonon peak shifts, expected for local heating, have been observed (see experimental data in the Supplemental Information). We have previously detected local heating

effects with Raman spectroscopy in a range of materials and different device structures, and are confident in the accuracy of such a method^{46,47}.”

Figure 1 d) is a standard plot for the CDW data, which illustrates the resistance change due to the depinning of CDW. **It does not show any heating effect at the used voltages** (below 1 V) and current (micro-Amperes) levels. The drop in the resistance is visible (it is not abrupt) and consistent with I-Vs. Moreover, the nonlinearity in the differential resistance vanishes as the temperature rises above the CDW transition temperature (see Figure S3c), further confirming that the observed nonlinear transport is intrinsic to CDW behavior. Additionally, the temperature-dependent shift in the threshold field, V_t , is a well-established characteristic of CDW systems and has been reported in various quasi-one-dimensional materials. We observe similar behavior in our NbS₃ devices (Fig. 6c), supporting the CDW origin of the threshold phenomena. In the prior studies with NbS₃, we also confirmed it with Shapiro step measurements (M. Taheri, N. Sesing, T. T. Salguero, and A. A. Balandin. “Electric-field modulation of the charge-density-wave quantum condensate in h-BN/NbS₃ quasi-2D/1D heterostructure devices.” *Appl. Phys. Lett.*, 123, 233101 (2023)).

- While data in Fig.1d concern the small values of V ($V < 0,2V$) those in Fig.1e are extended to 1V. How are the dV/dI characteristics till 1V in order to see possible jumps or inhomogeneities masked in the log-log I-V curves in fig.1e ?.

RESPONSE: The differential resistance vs. voltage plots in Figure 1d are presented on a semi-logarithmic scale to clearly illustrate the resistance behavior under both positive and negative bias polarities. This data presentation style is commonly used in the CDW research community to highlight symmetric or homogeneous changes in resistance across both bias directions. Regarding the reviewer’s question about the limited voltage range in Figure 1d, we intentionally restricted the bias to lower voltages because the primary objective of these measurements was to investigate the behavior of the threshold field, V_t . In contrast, during the noise measurements, we simultaneously

recorded the transport characteristics over an extended bias range (~ 1 V). We have provided the full transport behavior plots in the Supplemental file for the three devices presented in the manuscript. Additionally, we updated Fig 1d with a new figure showing data at more temperatures and higher bias ranges.

-Data in Fig1e show that the CDW current (evaluated from the extended (dashed line) Ohmic law (without taking care of any thermal effect) follows a power law without saturation till more than 10 times the threshold value. How to reconcile this result with theories which evaluate the high field limit of the CDW conductance?.

RESPONSE: There were no self-heating effects at the examined bias ranges

(please see the response above, information added to the main text and Supplementary). Indeed, the CDW current continues to increase in $(\text{TaSe}_4)_2\text{I}$ nanowires without saturation in the examined bias voltage range. In contrast, in the NbS_3 nanowires, the current shows the signs of saturation. In $(\text{TaSe}_4)_2\text{I}$ nanowires, the current will likely saturate at biases higher than what we examined, but we do not want to bias our devices too high to avoid possible self-heating (mentioned by the Reviewer). The fact that $(\text{TaSe}_4)_2\text{I}$ is a “special” CDW material was noticed early, starting from Gruner’s work at UCLA. The CDW transport in this material was described by underdamped oscillator. It appears that the CDW damping is weaker in this material. We have added some considerations about the specifics of this material. However, no microscopic transport theory exists to conclude. We made an important contribution toward it by developing the first comprehensive phenomenological noise theory in CDW conductors with low damping that the Reviewer can probably appreciate. The following sections have been added.

“We now address another fundamental question: Is the ultra-low noise of sliding condensate unique to the IC-CDW phase of $(\text{TaSe}_4)_2\text{I}$, or is it a property of many quasi-1D CDW conductors? The early studies of the “broadband” noise in bulk CDW crystals focused on the threshold noise^{15–17} and never considered the possibility of noise reduction below the normal electron limit. A few recent studies for TaS_3 and NbSe_3 found that beyond V_t , the noise returns to the normal electron

limit^{37,48}, in contrast to our observation of continuous noise reduction in the sliding regime of (TaSe₄)₂I devices. Other potential candidates for ultra-low noise transport should include fully-gapped CDW materials where the collective, strongly-correlated current of the sliding electron-lattice condensate is dominant. We experimented with nanowires of NbS₃-II, which exists in the CDW phase at RT³². The NbS₃-II nanowire device fabrication and measurements followed the same protocols as with (TaSe₄)₂I nanowire devices. Figure 6 (a-f) shows transport and noise characteristics for a representative NbS₃-II nanowire device. The most important observation is that the normalized spectral density in the NbS₃-II device reduces below the normal electron limit, similar to that in the (TaSe₄)₂I devices. The noise spectral density scaling with current is also consistent, *i.e.*, noise *reduces* with increasing current. It reveals a similar, $S_I/I^2 \propto 1/I$, dependence for the current above 10 μ A. We note that the noise reduction in NbS₃-II is achieved at temperatures above RT. The latter is important for practical applications, *i.e.* in interconnects. There is one difference in the noise behavior in the two materials: it appears the noise spectral density in NbS₃-II nanowires shows saturation behavior, even though the noise level is below the normal electron limit. This suggests that we may be reaching the intrinsic electron-lattice condensate noise in NbS₃-II nanowires but not in (TaSe₄)₂I nanowire devices (see additional data in Supplementary Materials).

The origin of the ultra-low-noise collective current in (TaSe₄)₂I nanowire, and the lack of noise saturation, may be related to the fact that the CDW transport in this specific material is described by the *under-damped* oscillator model^{49,50}. Compared to other materials like NbSe₃ and *o*-TaS₃, (TaSe₄)₂I has the largest energy gap and heaviest CDW effective mass, $m^*/m \approx 10^4$, yielding the lowest CDW damping. The non-linear CDW conductivity in this material continues to increase rather than saturate due to damping, in the examined bias voltages. The current in NbS₃-II shows signs of saturation at higher bias voltages (see Figure 6 (b)), indicating the onset of stronger CDW current damping, which likely translates to higher intrinsic CDW noise and eventual flattening of the total noise level (see Figure 6 (e) and (f)). Based on the recent realization that (TaSe₄)₂I is a Weyl semimetal^{30,31}, one may also argue that topological scattering suppression has relevance to the noiseless current of the sliding condensate in this material.”

-Eq. 2 is the combination of Eq.(5.6) and 5.14) from Ref.1

The first term results from the “single particle” model – classical particle moving in a sinusoidal washboard potential -in the case where the voltage is regulated yields $j \propto (E-E_t)^{1/2}$, (however in the part but” Electronic transport and noise measurement in the (TaSe4)2I nanowires” it seems that measurements were performed with a regulated current. In that case the threshold is manifested by an infinite negative differential resistance that is not the case in Fig.1d).

The second term provides from a different model due to dynamical critical phenomenon with the CDW velocity or the CDW current is a power law $(E-E_t)$ with an exponent $3/2$ to be compared with $1/2$ in the single particle model (however see \square and $2\square$ in eq.2). No real justification is made for the addition of two different threshold current with non-linear voltages with different exponents, even a third one - see figure caption of Extended data Fig.6).

RESPONSE:

There are several points here. The first is about writing the non-linear current $I_c = I_{c1} + I_{c2} = \Gamma_c [(V^2 - V_t^2)^a + (V - V_t)^{2a}] \theta(V - V_t)$ as a linear combination of the two non-linear current terms: $I_{c1} = \Gamma_c (V^2 - V_t^2)^a \theta(V - V_t)$ and $I_{c2} = \Gamma_c (V - V_t)^{2a} \theta(V - V_t)$. These two equations are based on the two equations noted above but combined in a way to minimize the number of free parameters. Thus, there is only one unknown exponent a , one proportionality constant Γ_c , and one threshold voltage V_t . The threshold voltage is pretty obvious from the data, so there are really just two fitting parameters a and Γ_c . The value for a for 2 different samples of different geometries is 0.9 with all parameters listed in the table on p. 16 of the Supplemental Material. While either I_{c1} or I_{c2} give a pretty good fit to the data, both are needed to match the derivative over the full bias range. Near V_t , I_{c1} gives rise to the small discontinuous turn-on feature right above V_t in Figs. 1e and 4e. This term also gives rise to the singular peak in the noise at V_t . The second term gives the $1/I$ scaling of S/I^2 at higher bias. Without this term, the noise would fall off even faster than $1/I$. The contributions of the individual terms in the

resulting noise expression (now Eq. (4) of the main text) are shown in Fig. S6 of the Supplement. Thus, the two non-linear current expressions with 2 free parameters are needed to fit both the current and the noise, and their underlying form is guided by the prior current expressions.

The second point concerns the biasing. The biasing is described on p. 4 of Methods. The devices are voltage biased – corresponding to the $E = \text{const}$ condition for which the $(E^2 - E_t^2)^{1/2}$ form is derived. The second form $(E - E_t)^\xi$ captures the larger biases well. Both are needed to match the current and the noise over the full bias range.

The final point is related to comment ‘even a third one - see figure caption of Extended data Fig.6’. I think there is possible confusion caused by bad notation here. There are only two components to the nonlinear current, I_{c1} and I_{c2} . Taking the derivative, squaring, and taking the ensemble averages, as now shown in Eqs. (3) and (4) of the main text (and also below), gives rise to four terms in the noise and three terms proportional to δV_t . These three terms were denoted as V_{t1} , V_{t2} , and V_{t3} in the original Extended Data Fig. 6. We now denote these three contributions resulting from fluctuations in the threshold voltage as S_{t1}/I^2 , S_{t2}/I^2 , and S_{t3}/I^2 . We emphasize that there is only one threshold voltage and only two current components.

In the revised text, we added the following.

“The general phenomenological theory of noise in CDW condensate conductors

To further understand the unusual noise reduction in fully-gapped CDW nanowires, we developed a comprehensive phenomenological noise theory in non-linear approximation. It has been accomplished by extending the “two-fluid” approach to include explicitly the *non-linear* voltage dependence of the collective current in $(\text{TaSe}_4)_2\text{I}$. The total current, $I = I_n + I_c$, consists of a normal current $I_n = G_n V$, and the sliding CDW current modeled as⁴⁰:

$$I_c = I_{c1} + I_{c2} = \Gamma_c [(V^2 - V_t^2)^a + (V - V_t)^{2a}] \theta(V - V_t). \quad (2)$$

Here, $\theta(V - V_t)$ is the unit step function that ensures the CDW contribution begins only after depinning. We denote the two components of the collective current with different non-linear voltage dependencies as $I_{c_1} = \Gamma_c(V^2 - V_t^2)^a \theta(V - V_t)$ and $I_{c_2} = \Gamma_c(V - V_t)^{2a} \theta(V - V_t)$, and Γ_c is a generalized conductance relating the current to the non-linear voltage. The transport behavior of our device exhibits a distinct kink at V_t . Similar changes in the bias dependence near threshold were previously attributed to the CDW creep before the onset of condensate sliding⁴¹⁻⁴³. Our phenomenological model incorporates two nonlinear terms with distinct power dependencies to account for the initial depinning dynamics and the subsequent sliding behavior of the CDW beyond the threshold. The accuracy of the description is confirmed by the excellent fit to the experimental data of all examined devices (Figure 4 (e) and Supplementary Materials).

Assuming uncorrelated, zero-mean fluctuations in the physical parameters, G_n , Γ_c , and V_t , the total first-order fluctuation in current can be expressed as:

$$\begin{aligned} \delta I &= \frac{\partial I}{\partial G_n} \delta G_n + \frac{\partial I}{\partial \Gamma_c} \delta \Gamma_c + \frac{\partial I}{\partial V_t} \delta V_t \\ &= \frac{V \delta G_n}{G_n} + \left\{ \frac{\Gamma_c [(V^2 - V_t^2)^a + (V - V_t)^{2a}] \delta \Gamma_c}{\Gamma_c} - \frac{2a \Gamma_c V_t [V_t^2 (V^2 - V_t^2)^{(a-1)} + V_t (V - V_t)^{(2a-1)}] \delta V_t}{V_t} \right\} \theta(V - V_t). \end{aligned} \quad (3)$$

Dividing by the total current, squaring, and taking the ensemble average, we obtain the normalized noise power expressed as:

$$\begin{aligned} \frac{\langle \delta I^2 \rangle}{I^2} &= \frac{\langle \delta G_n^2 \rangle}{G_n^2} \left(\frac{I_n}{I} \right)^2 + \left\{ \frac{\langle \delta \Gamma_c^2 \rangle}{\Gamma_c^2} \left(\frac{I_c}{I} \right)^2 + \frac{\langle \delta V_t^2 \rangle}{V_t^2} \left[\frac{2a \Gamma_c \{ V_t^2 (V^2 - V_t^2)^{(a-1)} + V_t (V - V_t)^{(2a-1)} \}}{I} \right]^2 \right\} \theta(V - V_t) \\ &= \frac{\langle \delta G_n^2 \rangle}{G_n^2} \frac{I_n^2}{I^2} + \left\{ \frac{\langle \delta \Gamma_c^2 \rangle}{\Gamma_c^2} \frac{I_c^2}{I^2} + 4a^2 \left[\frac{V_t^4}{(V^2 - V_t^2)^2} \frac{I_{c_1}^2}{I^2} + \frac{V_t^2}{(V - V_t)^2} \frac{I_{c_2}^2}{I^2} + \frac{2V_t^3}{(V^2 - V_t^2)(V - V_t)} \frac{I_{c_1} I_{c_2}}{I^2} \right] \frac{\langle \delta V_t^2 \rangle}{V_t^2} \right\} \theta(V - V_t). \end{aligned} \quad (4)$$

The first term is the noise of the normal current, the second term is the noise due to fluctuations of the generalized CDW conductance, Γ_c , and the third term is the noise due to fluctuations in the threshold voltage, V_t . At large voltages ($V \gg V_t$), the terms proportional to $\langle \delta V_t^2 \rangle / V_t^2$ are suppressed, and Eq. (4) reduces to a form similar to Eq. (1). At $V = V_t^+$, the normalized noise due to fluctuations in V_t is singular for $a < 1$, but in the limit of large $V \gg V_t$, it falls off rapidly.

Let us look closer at Figure 4 (f), which shows the noise calculated from Eq. (4) superimposed on the experimental data. The fitting to the experimental I-Vs (see Figure 4 (e)), allows one to extract the parameters, which are then used as inputs to the noise model. The first observation is that our theory accurately describes the noise peak due to fluctuations in V_t and how it rolls off as $1/I$. The bias dependence of the weighted normal electron noise and the CDW condensate noise are described as $[(\langle \delta G_n^2 \rangle / G_n^2)(I_n^2 / I^2)]$ and $[(\langle \delta \Gamma_c^2 \rangle / \Gamma_c^2)(I_c^2 / I^2)]$, respectively. The noise of the normal electrons dominates below the threshold voltage, V_t . Both noise components are constant at lower and higher bias regimes, when $I \approx I_n$ and $I \approx I_c$, respectively. In the high-bias regime where CDW noise emerges, it can saturate the total noise if its amplitudes grow large enough. The bias dependence of the threshold field fluctuations, $\langle \delta V_t^2 \rangle / V_t^2$, becomes pronounced near V_t . The total contribution from the threshold fluctuations to the noise contains the three distinct components within the square brackets of Eq. (4), each with a characteristic bias dependence. The first one, $S_{t_1} / I^2 = \{4a^2 V_t^4 / (V^2 - V_t^2)^2\} (I_{c_1}^2 / I^2) (\langle \delta V_t^2 \rangle / V_t^2)$ dominates near threshold and produces a noise peak that rapidly drops as $1/I^2$. The second one, $S_{t_2} / I^2 = \{4a^2 V_t^2 / (V^2 - V_t^2)^2\} (I_{c_2}^2 / I^2) (\langle \delta V_t^2 \rangle / V_t^2)$ dominates at $V \gg V_t$, resulting in noise scaling as $1/I$. The third one, $S_{t_3} / I^2 = [8a^2 V_t^3 / \{(V^2 - V_t^2)(V - V_t)\}] \{(I_{c_1} I_{c_2}) / I^2\} (\langle \delta V_t^2 \rangle / V_t^2)$ drops approximately as $1/I^{1.5}$ in the intermediate regime. Together, these components capture accurately the evolution of the noise across the depinning transition and into the CDW sliding regime. The details of the model fitting and resulting noise behavior across different bias regimes are presented in the Supplemental Materials.”

From their Eq.2 the authors derive the noise spectral density formed of 3 terms: noise of the normal current, noise of the fluctuations of the threshold voltage (near V_t) and fluctuations of the CDW conductance. They found that the noise spectra can be accurately fitted with only the noise amplitude of the normal current and that of threshold fluctuations but consequently without noise for the CDW conductance. However the values of different parameters such these three values of V_t are not indicated.

RESPONSE: In the original submission, the values of the fitting were indicated in the Extended Data Figure 6 (see the labels in the figures with noise amplitudes and captions with the description). In the revised manuscript, we provided all the details of the theory and fitting (see the main text and Supplementary Materials sections). We also included another material system to add support to our claims. All derivation details of the theory were provided in the Supplement (see below).

“The phenomenological theory of 1/f noise in non-linear approximation

We now derive the general phenomenological theory of 1/f noise in CDW conductors using a non-linear expression for current that captures transport characteristics of CDW conductors with low damping. The initial expression for the current can be chosen to match the specific material of interest closely. Here, we start with the equation for (TaSe₄)₂I, a Weyl CDW conductor with low damping. The total current can be best described by collective current with two non-linear components:

$$I = I_n + I_c = I_n + (I_{c_1} + I_{c_2}) = G_n V + \Gamma_c [(V^2 - V_t^2)^a + (V - V_t)^{2a}] \theta(V - V_t) \quad (S1)$$

Here, the exponent $\alpha = 2a$, I_{c_1} and I_{c_2} represents two collective current components with different non-linear voltage dependencies.

Let us assume uncorrelated, zero-mean fluctuations for, $\delta G_n, \delta \Gamma_c, \delta V_t$. Then

$$\delta I = \frac{\partial I}{\partial G_n} \delta G_n + \frac{\partial I}{\partial \Gamma_c} \delta \Gamma_c + \frac{\partial I}{\partial V_t} \delta V_t \quad (S2)$$

$$\frac{\partial I}{\partial G_n} = V \quad (S3)$$

$$\frac{\partial I}{\partial \Gamma_c} = (V^2 - V_t^2)^a + (V - V_t)^{2a} \quad (S4)$$

$$\frac{\partial I}{\partial V_t} = -2a \Gamma_c [V_t (V^2 - V_t^2)^{(a-1)} + (V - V_t)^{(2a-1)}] \quad (S5)$$

We can write the current fluctuation as:

$$\delta I = V \delta G_n + [(V^2 - V_t^2)^a + (V - V_t)^{2a}] \delta \Gamma_c - 2a \Gamma_c [V_t (V^2 - V_t^2)^{(a-1)} + (V - V_t)^{(2a-1)}] \delta V_t \quad (S6)$$

$$\delta I = \frac{V\delta G_n}{G_n} + \frac{\Gamma_c[(V^2 - V_t^2)^a + (V - V_t)^{2a}]\delta\Gamma_c}{\Gamma_c} - 2a\Gamma_c \frac{V_t[V_t^2(V^2 - V_t^2)^{(a-1)} + V_t(V - V_t)^{(2a-1)}]\delta V_t}{V_t} \quad (S7)$$

$$\delta I = \delta G_n \frac{VG_n}{G_n} + \delta\Gamma_c \frac{\Gamma_c[(V^2 - V_t^2)^a + (V - V_t)^{2a}]}{\Gamma_c} - \frac{2a\Gamma_c[V_t^2(V^2 - V_t^2)^{(a-1)} + V_t(V - V_t)^{(2a-1)}]\delta V_t}{V_t} \quad (S8)$$

$$\delta I = \frac{\delta G_n}{G_n} I_n + \frac{\delta\Gamma_c}{\Gamma_c} I_c - \left[\delta V_t \times 2a\Gamma_c \frac{\{V_t^2(V^2 - V_t^2)^{(a-1)} + V_t(V - V_t)^{(2a-1)}\}}{V_t} \right] \quad (S9)$$

Dividing by the total current, we obtain:

$$\frac{\delta I}{I} = \left(\frac{\delta G_n}{G_n} \right) \left(\frac{I_n}{I} \right) + \left(\frac{\delta\Gamma_c}{\Gamma_c} \right) \left(\frac{I_c}{I} \right) - \left(\frac{\delta V_t}{V_t} \right) \left[2a\Gamma_c \frac{\{V_t^2(V^2 - V_t^2)^{(a-1)} + V_t(V - V_t)^{(2a-1)}\}}{I} \right] \quad (S10)$$

As the next step, we square, take the ensemble average, and assume no cross correlation to get to the final expression for noise in this non-linear model:

$$\frac{\langle \delta I^2 \rangle}{I^2} = \frac{\langle \delta G_n^2 \rangle}{G_n^2} \left(\frac{I_n}{I} \right)^2 + \frac{\langle \delta\Gamma_c^2 \rangle}{\Gamma_c^2} \left(\frac{I_c}{I} \right)^2 + \frac{\langle \delta V_t^2 \rangle}{V_t^2} \left[2a\Gamma_c \frac{\{V_t^2(V^2 - V_t^2)^{(a-1)} + V_t(V - V_t)^{(2a-1)}\}}{I} \right]^2 \quad (S11)$$

$$\frac{\langle \delta I^2 \rangle}{I^2} = \frac{\langle \delta G_n^2 \rangle}{G_n^2} \left(\frac{I_n}{I} \right)^2 + \frac{\langle \delta\Gamma_c^2 \rangle}{\Gamma_c^2} \left(\frac{I_c}{I} \right)^2 + \frac{\langle \delta V_t^2 \rangle}{V_t^2} 4a^2 \Gamma_c^2 \frac{[V_t^2(V^2 - V_t^2)^{(a-1)} + V_t(V - V_t)^{(2a-1)}]^2}{I^2} \quad (S12)$$

$$\frac{\langle \delta I^2 \rangle}{I^2} = \frac{\langle \delta G_n^2 \rangle}{G_n^2} \left(\frac{I_n}{I} \right)^2 + \frac{\langle \delta\Gamma_c^2 \rangle}{\Gamma_c^2} \left(\frac{I_c}{I} \right)^2 + \frac{\langle \delta V_t^2 \rangle}{V_t^2} \times f(V, V_t) \quad (S13)$$

Where,

$$f(V, V_t) = 4a^2 \Gamma_c^2 \frac{[V_t^2(V^2 - V_t^2)^{(a-1)} + V_t(V - V_t)^{(2a-1)}]^2}{I^2} \quad (S14)$$

$$f(V, V_t) = 4a^2 \Gamma_c^2 \frac{[V_t^4(V^2 - V_t^2)^{2(a-1)} + V_t^2(V - V_t)^{2(2a-1)} + 2V_t^3(V^2 - V_t^2)^{(a-1)}(V - V_t)^{(2a-1)}]}{I^2} \quad (S15)$$

$$f(V, V_t) = 4a^2 \left[V_t^4 \frac{\Gamma_c^2(V^2 - V_t^2)^{2(a-1)}}{I^2} + V_t^2 \frac{\Gamma_c^2(V - V_t)^{2(2a-1)}}{I^2} + 2V_t^3 \frac{\Gamma_c^2(V^2 - V_t^2)^{(a-1)}(V - V_t)^{2(2a-1)}}{I^2} \right] \quad (S16)$$

$$f(V, V_t) = 4a^2 \left[V_t^4 \frac{\Gamma_c^2(V^2 - V_t^2)^{2a}}{(V^2 - V_t^2)^2 I^2} + V_t^2 \frac{\Gamma_c^2(V - V_t)^{4a}}{(V - V_t)^2 I^2} + 2V_t^3 \frac{\Gamma_c^2(V^2 - V_t^2)^{2a}(V - V_t)^{4a}}{(V^2 - V_t^2)(V - V_t) I^2} \right] \quad (S17)$$

$$f(V, V_t) = 4a^2 \left[V_t^4 \frac{\{\Gamma_c(V^2 - V_t^2)^a\}^2}{(V^2 - V_t^2)^2 I^2} + V_t^2 \frac{\{\Gamma_c(V - V_t)^{2a}\}^2}{(V - V_t)^2 I^2} + 2V_t^3 \frac{\Gamma_c\{(V^2 - V_t^2)^a\} \times \Gamma_c\{(V - V_t)^{2a}\}^2}{(V^2 - V_t^2)(V - V_t) I^2} \right] \quad (S18)$$

$$f(V, V_t) = 4a^2 \left[V_t^4 \frac{I_{c_1}^2}{(V^2 - V_t^2)^2 I^2} + V_t^2 \frac{I_{c_2}^2}{(V - V_t)^2 I^2} + 2V_t^3 \frac{I_{c_1} I_{c_2}}{(V^2 - V_t^2)(V - V_t) I^2} \right] \quad (\text{S19})$$

$$f(V, V_t) = 4a^2 \left[\frac{V_t^4}{(V^2 - V_t^2)^2} \frac{I_{c_1}^2}{I^2} + \frac{V_t^2}{(V - V_t)^2} \frac{I_{c_2}^2}{I^2} + \frac{2V_t^3}{(V^2 - V_t^2)(V - V_t)} \frac{I_{c_1} I_{c_2}}{I^2} \right] \quad (\text{S20})$$

By adding Eq. (M20) in Eq. (M13), the total noise fluctuation becomes,

$$\frac{\langle \delta I^2 \rangle}{I^2} = \frac{\langle \delta G_n^2 \rangle}{G_n^2} \left(\frac{I_n}{I} \right)^2 + \frac{\langle \delta \Gamma_c^2 \rangle}{\Gamma_c^2} \left(\frac{I_c}{I} \right)^2 + \frac{\langle \delta V_t^2 \rangle}{V_t^2} 4a^2 \left[\frac{V_t^4}{(V^2 - V_t^2)^2} \frac{I_{c_1}^2}{I^2} + \frac{V_t^2}{(V - V_t)^2} \frac{I_{c_2}^2}{I^2} + \frac{2V_t^3}{(V^2 - V_t^2)(V - V_t)} \frac{I_{c_1} I_{c_2}}{I^2} \right] \theta(V - V_t) \quad (\text{S21})$$

This represents the noise model equation for (TaSe₄)₂I device, where,

$$\frac{S_{t_1}}{I^2} = 4a^2 \frac{V_t^4}{(V^2 - V_t^2)^2} \frac{I_{c_1}^2}{I^2} \frac{\langle \delta V_t^2 \rangle}{V_t^2} \quad (\text{S22})$$

$$\frac{S_{t_2}}{I^2} = 4a^2 \frac{V_t^2}{(V - V_t)^2} \frac{I_{c_2}^2}{I^2} \frac{\langle \delta V_t^2 \rangle}{V_t^2} \quad (\text{S23})$$

$$\frac{S_{t_3}}{I^2} = 4a^2 \frac{2V_t^3}{(V^2 - V_t^2)(V - V_t)} \frac{I_{c_1} I_{c_2}}{I^2} \frac{\langle \delta V_t^2 \rangle}{V_t^2} \quad (\text{S24})$$

At $V = V_t^+$, the normalized noise due to fluctuations in V_t is singular for $a < 1$, but in the limit of large $V \gg V_t$, it falls off rapidly.

At $V \gg V_t$,

$\frac{S_{t_1}}{I^2}, \frac{S_{t_2}}{I^2}$ & $\frac{S_{t_3}}{I^2}$ falls off as $\frac{1}{V^4}, \frac{1}{V^2}$ & $\frac{1}{V^3}$ respectively, which corresponds to $\frac{1}{I^2}, \frac{1}{I}$ & $\frac{1}{I^{1.5}}$.

Fitting parameters from the I-V model

The I-V model equation is fitted with experimental data. The obtained fitting parameters are as follows:

Device	$G_n(S)$	Γ_c	$V_t (V)$	a
1	1.947×10^{-6}	6.72×10^{-6}	0.08241	0.9

2	3.912×10^{-6}	3.556×10^{-6}	0.04751	0.89393
3	1.638×10^{-6}	4.31×10^{-6}	0.04392	0.79445

Fitting parameters for the noise model

To fit the noise model equation with the experimental data, we plugged in values of the I-V fitted parameters (G_n, Γ_c, V_t & a). The noise amplitudes are not calculated from the first principles but rather determined by comparing the experimental data with the model. For that, we calculated the average constant noise level, S_I/I^2 , of the normal carrier noise from the experimental data since $S_I/I^2 = \text{constant}$. The average noise is taken as the value of the linear noise amplitude, $\langle \delta G_n^2 \rangle / G_n^2$. The CDW noise amplitude, $\langle \delta \Gamma_c^2 \rangle / \Gamma_c^2$ cannot be determined accurately since the experimental noise never saturates. We took the upper-bound value before the noise could be saturated at higher biases. For the threshold field noise amplitude, $\langle \delta V_t^2 \rangle / V_t^2$, we plugged in different values and superimposed the noise model output with the experimental data. The threshold amplitude was determined based on the best fit of the model with the experimental data at higher biases, where the noise follows, $S_I/I^2 \propto 1/I$.

The table contains the values used for the fluctuation terms to fit the noise model.

Device	$\frac{\langle \delta G_n^2 \rangle}{G_n^2}$	$\frac{\langle \delta \Gamma_c^2 \rangle}{\Gamma_c^2}$	$\frac{\langle \delta V_t^2 \rangle}{V_t^2}$
1	4.9×10^{-9}	1.0×10^{-12}	3.5×10^{-7}
2	9.5×10^{-9}	1.0×10^{-12}	1.9×10^{-6}
3	1.3×10^{-9}	1.0×10^{-12}	1.45×10^{-6}

Calculation of the transport parameters and weighted noise contributions

In the incommensurate phase, total device current combines linear current and CDW current, such that: $I = I_n + I_c$, where, I_n is normal current, I_c is the CDW current, G is the total conductance,

which combines linear and CDW terms, such as, $G = G_n + G_c$. Here, G_n is normal carrier conduction, which is calculated as the slope of I-V data in the linear regime, such as, $G_n = \frac{dI_n}{dV}$ and $G_c = \frac{dI}{dV} - G_n$, is the CDW carrier conductance, where $G = \frac{dI}{dV}$. The CDW current is calculated as, $I_c = I - I_n = I - VG_n$. We now derive Eq. (1) of the main text. In the two-fluid model, total device current is given by the equation: $I = VG = I_n + I_c$. If the fluctuations in the two conductivities are independent and the cross-correlation is neglected, we get the total current fluctuations as:

$$\delta I = \delta I_n + \delta I_c = V(\delta G_n + \delta G_c). \quad (S1)$$

Taking the square of the fluctuation terms and neglecting the cross-correlated term, we have:

$$\delta I^2 = V^2 \delta G_n^2 + V^2 \delta G_c^2. \quad (S2)$$

Taking the ensemble average, we obtain:

$$\langle \delta I^2 \rangle = V^2 \langle \delta G_n^2 \rangle + V^2 \langle \delta G_c^2 \rangle. \quad (S3)$$

Writing in terms of spectral density for fluctuations, we get:

$$S_I = V^2 S_{G_n} + V^2 S_{G_c}. \quad (S4)$$

Dividing by the square of I , we write:

$$\frac{S_I}{I^2} = S_{G_n} \frac{V^2}{I^2} + S_{G_c} \frac{V^2}{I^2}, \quad (S5)$$

$$\frac{S_I}{I^2} = S_{G_n} \frac{1}{G^2} + S_{G_c} \frac{1}{G^2}. \quad (S6)$$

Therefore, the total current fluctuations, can be written as:

$$\frac{S_I}{I^2} = \frac{S_{G_n}}{G_n^2} \frac{G_n^2}{(G_n+G_c)^2} + \frac{S_{G_c}}{G_c^2} \frac{G_c^2}{(G_n+G_c)^2}. \quad (S7)$$

Here, $\frac{S_I}{I^2}$ is the total normalized current noise; $\frac{G_n^2}{(G_n+G_c)^2}$, $\frac{G_c^2}{(G_n+G_c)^2}$ are the normalized conductions;

$\frac{S_{G_n}}{G_n^2}$, $\frac{S_{G_c}}{G_c^2}$ are the normal and the CDW carrier conduction noise components; $\frac{S_{G_n}}{G_n^2} \frac{G_n^2}{(G_n+G_c)^2}$,

$\frac{S_{G_c}}{G_c^2} \frac{G_c^2}{(G_n+G_c)^2}$ are the weighted contributions of the normal and CDW carrier conduction noise components.”

I have taken consideration of this work but I consider that the conclusion of a CDW condensate moving without noise is not justified based on the presented data. There are two many uncertainties either in the experimental measurements and in the

theoretical and fitting process which raise some, in my sense, reasonable doubts on the conclusions.

RESPONSE: We have added extra data and explanations to **remove any doubts** about our claims. We are ready to demonstrate the measurement procedures in the laboratory if needed.

Thus I will not recommend publication of this work in Nature Communications

RESPONSE: We sincerely hope that Reviewer 2 will change his/her opinion after considering all the evidence we provided and additional data, including for a different material.

Reviewer #3 (Remarks to the Author):

RESPONSE: We sincerely thank the co-reviewer for their valuable time and constructive feedback, which greatly helped improve our manuscript.

Reviewer #4 (Remarks to the Author):

The manuscript NCOMMS-25-46228-T by Subhajit Ghosh et al. describes a study of low-frequency electronic noise suppression in quasi-one-dimensional charge-density-wave (CDW) systems, specifically in $(\text{TaSe}_4)_2\text{I}$ nanowires, under conditions of sliding Frohlich condensate current.

The reported suppression of low-frequency $1/f$ noise in the sliding CDW regime is highly unusual and underlines a significant difference from conventional transport behavior in resistive materials. In this respect, the linear decrease of normalized noise spectral density, as opposed to the typical constant value in normal conductors, suggests a fundamentally different noise mechanism linked to collective transport. This is an indicator of new physics in the correlated transport regime. Moreover, the connection to real-world technological relevance, emphasizing technological relevance to quantum devices, low-noise communication, and sensing, strengthens the case for broad impact.

RESPONSE: We thank the Reviewer for stating that our work “suggests a fundamentally different noise mechanism linked to collective transport,” and that our study reports “new physics in the correlated transport regime,” and that our work has a broad impact. We take these comments as indications of the suitability of our paper for Nature Communications.

In the context of existing literature, the submitted work appears scientifically novel in its experimental observation of sub-thermal electronic noise in the sliding CDW regime, a behavior not established in previous CDW literature where broadband $1/f$ noise has been usually interpreted as a consequence of CDW dynamics near depinning. Moreover, recent studies, exploring CDW systems in reduced dimensions (nanowires, thin films), are concerned with transport phenomena but not noise suppression, as the case of the article here considered.

RESPONSE: We thank the Reviewer for recognizing the novelty of our work.

Although the work appears sufficiently novel and innovative for publication in Nature Communications, two important points deserve more clarification and should be strongly taken into account before a resubmission.

(l) The first point concerns the fact that the notion of using strong correlations to suppress noise, especially leveraging a collective mode like Frohlich conduction, is still underexplored and seems to be largely speculative. The authors need to better clarify this aspect, demonstrating that the theory developed is based on solid principles and the results obtained are not just material-specific observations. In this respect, further experimental investigations showing an analogous noise behavior as the one here reported, performed on different systems undergoing a similar CDW regime, could be very useful to address such a criticism.

RESPONSE: Following the Reviewer 3 request, we performed noise measurements on another interesting system, NbS₃, which reveals CDW phases at RT. We planned it as a separate publication, but based on the Reviewer's comment, we included it in this paper. This material system is also a fully gapped CDW where one can achieve dominance of the current by the sliding CDW condensate. We observed a similar suppression below the normal electron limit. However, there is one interesting difference: we can see some noise saturation below the normal electron limit. Below is a section added to the revised manuscript.

“The electronic noise of the CDW condensate in NbS₃ nanowires

We now address another fundamental question: Is the ultra-low noise of sliding condensate unique to the IC-CDW phase of (TaSe₄)₂I, or is it a property of many quasi-1D CDW conductors? The early studies of the “broadband” noise in bulk CDW crystals focused on the threshold noise^{15–17} and never considered the possibility of noise reduction below the normal electron limit. A few recent studies for TaS₃ and NbSe₃ found that beyond V_t , the noise returns to the normal electron limit^{37,48}, in contrast to our observation of continuous noise reduction in the sliding regime of

(TaSe₄)₂I devices. Other potential candidates for ultra-low noise transport should include fully-gapped CDW materials where the collective, strongly-correlated current of the sliding electron-lattice condensate is dominant. We experimented with nanowires of NbS₃-II, which exists in the CDW phase at RT³². The NbS₃-II nanowire device fabrication and measurements followed the same protocols as with (TaSe₄)₂I nanowire devices. Figure 6 (a-f) shows transport and noise characteristics for a representative NbS₃-II nanowire device. The most important observation is that the normalized spectral density in the NbS₃-II device reduces below the normal electron limit, similar to that in the (TaSe₄)₂I devices. The noise spectral density scaling with current is also consistent, *i.e.*, noise *reduces* with increasing current. It reveals a similar, $S_I/I^2 \propto 1/I$, dependence for the current above 10 μ A. We note that the noise reduction in NbS₃-II is achieved at temperatures above RT. The latter is important for practical applications, *i.e.* in interconnects. There is one difference in the noise behavior in the two materials: it appears the noise spectral density in NbS₃-II nanowires shows saturation behavior, even though the noise level is below the normal electron limit. This suggests that we may be reaching the intrinsic electron-lattice condensate noise in NbS₃-II nanowires but not in (TaSe₄)₂I nanowire devices (see additional data in Supplementary Materials).

The origin of the ultra-low-noise collective current in (TaSe₄)₂I nanowire, and the lack of noise saturation, may be related to the fact that the CDW transport in this specific material is described by the *under-damped* oscillator model^{49,50}. Compared to other materials like NbSe₃ and *o*-TaS₃, (TaSe₄)₂I has the largest energy gap and heaviest CDW effective mass, $m^*/m \approx 10^4$, yielding the lowest CDW damping. The non-linear CDW conductivity in this material continues to increase rather than saturate due to damping, in the examined bias voltages. The current in NbS₃-II shows signs of saturation at higher bias voltages (see Figure 6 (b)), indicating the onset of stronger CDW current damping, which likely translates to higher intrinsic CDW noise and eventual flattening of the total noise level (see Figure 6 (e) and (f)). Based on the recent realization that (TaSe₄)₂I is a Weyl semimetal^{30,31}, one may also argue that topological scattering suppression has relevance to the noiseless current of the sliding condensate in this material.”

(II) The second point, which is somewhat connected to the first one, regards an improvement of the experimental robustness and generality of the work, in order to elevate its impact from a specialized CDW study to a wider platform as, for example, for quantum/nanoelectronics.

RESPONSE: The general noise theory that we developed for CDW materials (something that has not been done after 50 years of studies) will have broad implications for quantum and nanoelectronics. In the revised manuscript and Supplemental Materials, we provided all details. In the original manuscript, we had a paragraph on benchmarking against conventional electronics, which we believe addresses another part of the Reviewer's question.

As a minor comment, due to the high visibility and high impact of Nature Communications journal, my suggestion to the authors is to mention other systems and materials where an evident reduction of the $1/f$ noise level has been observed. This is the case, for example, of low-dimensional systems and topological insulators. A detailed description of key similarities and key differences in noise suppression mechanisms could also be given.

RESPONSE: The type of noise suppression we observed and noise scaling with the current is unique **for fully gapped CDW materials** with relatively low damping. We proved it for two different materials. We are not aware of any other results like these for any other material systems. This confirms the novelty and suitability of our paper for a Nature family publication. In the revised manuscript, we added a paragraph that clarifies the significance of the observed noise scaling.

“The fact that we observed a different dependence on the normalized noise spectral density in a CDW conducting channel is of great importance. It is well-known that the noise level, *i.e.*, normalized noise spectral density, in passive resistors does not depend on current: $S_I/I^2 \propto \text{constant}$ (or $S_I \propto I^2$). This fundamental property originates in the linear response theory³⁸. In terms of physics, it means that the current does not drive the resistance fluctuations in the sample but rather

makes them visible via Ohm's law. The fact that $S_I \propto I^2$ allowed for the introduction of the phenomenological Hooge formula, $S_I/I^2 \sim \alpha_H/fN$, where α_H is the Hooge parameter, and N is the number of charge carriers in the resistor. There are known cases of deviation from $S_I/I^2 \propto \text{constant}$ dependence in specific electronic devices, like diodes³⁹. In this work, we report the first observation of as $S_I/I^2 \propto 1/I$ scaling of noise in a generic resistor with two metal contacts. This means that increasing the current density will simultaneously decrease the noise level, which can become a game-changing advantage for interconnect applications.”

For all the reasons specified above, in my opinion, the paper is not suitable for publication in Nature Communications in its present form. A major revision, by addressing all the points indicated, is needed before the manuscript can be reconsidered.

RESPONSE: We sincerely hope that Reviewer 3 can now recommend our manuscript for publication, considering all the evidence we provided and additional data, including for a different material (at the Reviewer 3 request).

RESPONSE TO THE REVIEWERS' COMMENTS

Reviewers' comments:

Reviewer #1 (Remarks to the Author):

The comments from my earlier review have been satisfactorily addressed. The revised manuscript and supplement information are vastly improved from the previous version.

Response: We thank Reviewer #1 for stating that we have satisfactorily addressed the comments from the previous submission. We appreciate Reviewer #1's statement that "revised manuscript and supplement information are vastly improved from the previous version".

My only comment is for the authors to consider adding their comments about the novelty of this work compared to previous publication (NbSe₃). They have provided the arguments in the response which I find to be satisfactory, but it will be helpful to have them prominently stated in the revised manuscript.

Response: As suggested by Reviewer #1, in the revised manuscript, we added several sentences clarifying the novelty and explaining the principal difference of the present work with NbSe₃. As we mentioned in our previous response, NbSe₃ is a CDW material, which is not fully gapped. For this reason, the current is always dominated by normal electrons with some contribution from CDWs. As a result, in NbSe₃, we have not observed the noise reduction below the normal electron noise limit. Our new results constitute a major step forward. The novelty includes: (1) clear demonstration of the noise below the normal electron noise limit in two fully-gapped CDW materials; (2) scaling of the CDW condensate noise with bias voltage, which is different from scaling of the noise of normal electrons; (3) comprehensive phenomenological theory of 1/f noise in CDW materials (something that has not been done before, after 50 years of investigations). In the revised manuscript, we added the following sentences.

It is interesting to compare the noise dependence on bias in (TaSe₄)₂I and NbS₃ nanowires with that in NbSe₃ nanowires³⁷. While the noise in (TaSe₄)₂I and NbS₃ nanowires decreases below the normal electron limit when the CDW sliding current component starts to dominate, the noise in NbSe₃ nanowires saturates at the normal electron noise level. This important difference is explained by the fact that NbSe₃ is not a fully-gapped CDW material. For this reason, even after CDW depinning, the current is dominated by the normal electron contribution. Correspondingly, the noise level does not decrease below the normal electron limit. This observation adds to the importance of the present finding of noise reduction and unusual

scaling with the current in nanowires made of fully-gapped CDW materials such as $(\text{TaSe}_4)_2\text{I}$ and NbS_3 .

With that, I recommend publication.

Response: We thank Reviewer #1 for recommending our paper for publication in Nature Communications.

Reviewer #2 (Remarks to the Author):

I acknowledge the authors for the improvements they made in the revised version of their paper including in particular data on the CDW compound NbS_3 with its Peierls transition temperature above room temperature and their phenomenological theory of $1/f$ noise in Supplementary.

Response: We thank Reviewer #2 for acknowledging the improvements we have made in the revised manuscript, specifically that we added data for a new material system (this required significant additional work) and developed a complete phenomenological theory of $1/f$ noise in CDW materials (something that researchers have not managed to do in more than 50 years of research). We respectfully note that the main elements of the theory were presented in the Methods section of the main text rather than the Supplement. The Supplement contains details of the calculation and data extraction for the model.

However I did not get convincing answers to the questions raised in my first report, specifically concerning heating effects.

The fact that no softening of the phonon modes in Raman experiments is, for the authors, the direct proof of the absence of Joule heating. But the more simple and direct proof is the electric field dependence of the normal resistance which in the pinned CDW state below the threshold should keep its normal value. The activated temperature dependence of the resistance in the normal state of these fully gapped CDW materials is a very sensitive probe of possible heating effects.

No explanation has been given on the electric field dependence of the differential resistance below threshold as shown in Fig1d in a semilog plot in the original manuscript. Opportunely the updated rescaled Fig1d in the revised manuscript has in some way erased this dependency.

However this effect seems to be less important in NbS₃ as seen in Fig6c (but data in a linear plot would be important).

This question is of a huge importance since the way to extract the CDW noise results from the subtraction from the total noise of independent noise from the normal current and noise from fluctuations in the threshold voltage. Any change in the normal current noise may affect the result.

It appears difficult to ascertain as the authors state that Joule heating does not play any role.

I keep the conclusions of my first report

Response: We take responsibility for not grasping what Reviewer 2 was finding problematic in the data. Our focus was on the larger bias data ($V > V_t$), where all the interesting phenomena were occurring, rather than on the low-bias data before the threshold. We now understand the concern of Reviewer 2 with respect to the low-bias curves. To address Reviewer #2's concerns, we revised our manuscript accordingly.

Below, we explain in detail that there was no self-heating in our experiments and sort out the confusion regarding Figure 1 (d). We understand now where the misunderstanding might have originated.

Ironically, Figure 1 (d) (in the original and resubmitted version) is not even used in the transport and noise data analysis. Figure 1 (d) was meant as preliminary material characterization showing that we do have the correct (TaSe₄)₂I material with the transition temperature and threshold voltage. The I-V and conductivity that were used in the noise data analysis were shown in Figures 1 (e) and (f), and similar corresponding figures for other devices. These I-Vs were taken during the noise measurements. We have clarified it in the revised manuscript.

Moreover, even if there were some local heating, it would not affect the conclusion of the paper on the noise reduction below the normal electron limit, which was experimentally demonstrated for many devices made of (TaSe₄)₂I and, at Reviewer #3-#4's request, for another material system, NbS₃-II. The possibility of having the total noise below the normal electron limit was also rigorously proven theoretically (it could have been a separate paper). The noise of CDWs was not determined by simply subtracting the noise at low bias but rather extracted via a rigorous procedure, which included the use of analytical fitting to the actual experimental data (please see the theory section of the paper).

We start by pointing out that we used very low bias voltages and current levels, insufficient for any local heating. There is a simple formula from Fourier's law that allows one to estimate the self-heating, $\Delta T = [\text{Power}] \times [\text{Thermal Resistance}]$. There is no possibility of having noticeable

heating effects in our devices at the bias voltages (0.1 V) and current levels (nano-Ampere or fraction of micro-Ampere) across the depinning that we used. In order to have noticeable self-heating, one needs to apply voltages above 10 V and have order of magnitude larger currents. We address this in detail below, including the approach proposed by Reviewer #2.

Let us explain the source of confusion, for which we take responsibility. This paper was originally submitted to Nature, and we had to put all data for the main message into three panels, which did not allow us to explain things easily and created issues with the original submission to Nature Communications (the manuscript was transferred “as is”). The first 4 panels of Figure 1 were meant for materials characterization. Those panels should have been a separate figure, but were combined with the measurement for Device 1. In Figure 2, we combined the final noise data for Devices 1 and 2 (last panel).

Here is the measurement protocol, a standard one in the noise community, that Subhajit Ghosh (the first author) used in this study.

He measured the I-Vs twice. First, he did a quick measurement of I-Vs to characterize the material and verify that the device was working fine. Such representative I-Vs were used in Figure 1 (d), original and resubmitted versions. Then he placed the device structure in the noise measurement setup and started biasing the device at different voltages (from low to high), and measured noise at every voltage. This gives us another version of I-V (similar to the first one, but the actual voltage bias points may be different). This I-V, from the second measurement, is used in the analysis because it is the most relevant, taken simultaneously with the noise measurements. There can be slight variations in the flatness of the curves because CDW depinning is never completely coherent. There might be small variations compared to the first I-V measurement because the time of measurement is slower (one has to measure noise for each voltage bias point). However, one can see in Figure 1 (f) that at low bias, the curve is nearly flat.

The noise data and analysis in Figure 2 (a – e) correspond to this device without any signs of heating. Some kinks and deviation from flatness may be related to incomplete coherence of depinned CDW (the process does not start in the entire sample at once), CDW creep before the full sliding, and numerical errors of differentiation of the I-V in Origin (for Figure 1 (d)). We measure I-Vs and differentiate them numerically in Origin without function smoothing. What matters for the noise extraction and analysis is seen below (from the original submission and resubmission).

The I-V at low bias is perfectly linear, indicating no heating.
 Heating effects are not possible at 0.01 V and micro-A.

This is conductance: nearly flat

This I-V is taken during the noise measurement and used
 in data extraction for each device

This is the figure from the original submission and
 resubmission: all is correct and no self-heating.

In the original submission, in Figure 1 (d), we used a representative I-V for one device with multiple top electrodes. Again, that was numerical differentiation of the first I-V measurement data conducted to check the material quality and device function. This data was for a device with multiple electrodes to verify that the measurements in a 4-probe configuration and a 2-probe configuration give similar results. The proof of the same I-Vs for 2-probe and 4-probe was placed in the Supplemental Figure 3 (a). Plotted on a log-linear scale, it gave an exaggerated impression of a bump at low bias. The format was used to place several curves for different temperatures on the same plot. The curves that had some bumping in the middle were at the lower temperatures, $T = 120$ K and below, where the samples were substantially more resistive (hundreds of $M\Omega$). The focus of the paper was on noise analysis at higher temperature, $T = 180$ K – 200 K, which is closer to RT and has more practical relevance. At this temperature, the resistance of the samples was significantly lower, as required for reliable noise measurements, and the low-bias curves were flat. The noise analysis for NbS_3 was for RT. The curves at $T = 280$ K – 320 K are flat.

After Reviewer #2's first report, we had to fabricate a new device structure (the old one had died) to add more dependencies to Figure 1 (d). It had a slightly longer channel, which explains the difference in V_t . We thought that the concern was about self-heating at the largest applied bias voltage, and did not pay much attention to near-zero voltage. We replaced Figure 1 (d) to show more curves, but have not changed anything with Figures 1 (e) or 1 (f), which were used for the noise data processing. We did not notice that the caption was not revised properly to indicate this change. Figure 1 has been corrected now: Figure 1 (d) is now for Device 1 (used in the actual noise measurements) and plotted in a format allowing for seeing the low bias region.

Nothing was changed in Figures 1 (e) and 1 (f), which were used in the noise data processing for the same Device 1.

Below, we plotted numerically differentiated I-Vs specifically for Device 1 on a different, linear-logarithmic scale to show the dependencies more clearly.

Here is the same in log-log scale.

There is no self-heating and little deviation from flat dependence at the low bias. There are a few well-known physical reasons to expect slight bending and kinks in the differential resistance of CDW nanowires below the threshold. The CDW sliding does not start simultaneously throughout the sample volume, and the onset of collective current is gradual. There is another reason – the CDW creep transport regime, which is the partial movement of CDW before the sliding. It can also result in bending of the differential I-Vs. Our plots look “textbook-like”, similar to the ones reported for bulk crystals, where self-heating was not possible due to the large sample size. Depending on the resistance values, the deviations from a perfectly flat conductivity can be seen as more or less pronounced. We got consistent results for other devices, and included Devices 2 and 3 in the revised submission.

To sum up, to address Reviewer #2’s concerns, we added an extra explanation of how I-Vs are measured (once for material characterization and device test, and the second during the noise measurements) and indicated the possible reasons for the derivative I-V deviations from perfectly flat dependence.

In response to Reviewer #2’s comments, in the revised manuscript, we added the sentences:

The dependence was obtained by numerical differentiation of measured I-Vs.

[...] The V_t value shows a non-monotonic temperature dependence consistent with trends reported for other quasi-1D CDW systems.^{1,3,4}. Slight deviations from the perfectly flat derivative characteristics at low bias and kinks are attributed to CDW creep before the depinning and not completely coherent CDW depinning and sliding.

[...] These characteristics are measured for the second time during the noise measurements. They are further used in the noise data analysis.

We now address the additional data that we included in the previous resubmission. We used Raman spectroscopy for temperature sensing. This is a conventional method, which one of us, Alexander Balandin, used in his prior Nature Communications publication: Z. Yan, G. Liu, J. M. Khan, and A. A. Balandin, “Graphene quilts for thermal management of high-power GaN transistors,” Nature Commun., 3, 827, 2012. Raman thermometry is an established and commonly accepted technique. We note that if there were self-heating effects, they would have appeared prominently in NbS₃ devices. There were none.

We also conducted analytical and COMSOL calculations of the self-heating. There was a new long paragraph added with this extra data in the resubmitted version. Neither the analytical model nor the detailed COMSOL simulation predicts any self-heating. One can understand why it is negligible in our case: we only use a bias of ~1 V and a current of ~1 micro-A in the sliding regime. In the low-voltage linear regime, the current is in the nano-A range with applied voltages of only a few mV. At such low levels, Joule heating is physically impossible, regardless of sample dimensions.

Here is what we wrote in the resubmitted manuscript.

“In the data analysis, it is also important to exclude local heating effects. No Joule self-heating effects are expected in our test structures at the small bias voltages, about ~1 V, and low

current range, about $\sim 1 \mu\text{A}$. One can readily estimate the temperature rise, ΔT , from an analytical formula derived from Fourier's law: $\Delta T = P \times R_T = I \times V \times (T_{ox}/K \times L \times W)$. Here, P is the dissipated power in the nanowire, R_T is the thermal resistance of the SiO_2 layer, T_{ox} is the thickness of the SiO_2 layer, K_{ox} is the thermal conductivity of SiO_2 , L is the length of the channel, and W is the width of the channel. The formula assumes that the bulk Si substrate is the thermal sink, and the SiO_2 layer with its low thermal conductivity is the thermal barrier. Using the typical values, *i.e.*, $I = 1 \mu\text{A}$, $V = 1 \text{ V}$, $T_{ox} = 300 \text{ nm}$, $K_{ox} = 1 \text{ W/mK}$, $L = 5 \mu\text{m}$, and $W = 300 \text{ nm}$, we obtain for the thermal resistance $R_T \sim 0.2 \text{ MK/W}$ and, correspondingly, for the temperature rise $\Delta T \sim 0.2 \text{ K}$. We can elaborate the estimate and consider the thermal boundary resistance, R_C , between the nanowire and SiO_2 layer. It is well known that it is primarily defined by the interface quality rather than specific channel material. Assuming the worst-case scenario of low interface conductance, $G_T = 15 \text{ MW/m}^2\text{K}^{44,45}$, we obtained an extra thermal resistance of $\sim 0.2 \text{ MK/W}$, which doubles the total resistance and temperature rise. On the other side, conduction to metal contacts along the channel may reduce the temperature rise. These simple but reliable estimates show that in our devices, in the considered bias and current ranges, the local heating is negligible. One has to pass currents in the mA range, a factor of $\times 1000$ larger, to induce significant Joule heating. The temperature rise was also assessed using COMSOL software tools for a given device structure and bias voltages. The maximum temperature increase at the hot spot (center of the channel) is approximately 1 K at a voltage drop of 0.9 V. The simulation results confirmed negligible self-heating effects (see Supplemental Materials)."

Here we address another Reviewer #2's comment: "The activated temperature dependence of the resistance in the normal state of these fully gapped CDW materials is a very sensitive probe of possible heating effects."

We assume Reviewer #2 meant "resistance of the normal current" rather than "resistance in the normal state", since in the "normal state", there is no gap.

Following the comment, we conducted additional analysis based on the low-bias ($V < V_t$) current in the fully gapped CDW phase. From the relationship $R_n \propto \exp(\Delta/kT)$, if $R_n(V)$ changes with voltage such that $R_1(V_1) < R_0(V = 0)$, then

$$T_1 = \left(\frac{1}{T_0} + \ln \left(\frac{R_1}{R_0} \right) \frac{1}{T_\Delta} \right)^{-1},$$

where $T_\Delta = \Delta/k_B$. We used this formula, assuming that the changes in the current below the threshold are related to self-heating (even though it is unlikely). The resulting ΔT was found to be negligible. In all checked cases, the temperature rise was below 1 K. The formula works below the CDW depinning.

We thank Reviewer #2 for suggesting this method for assessing the temperature. The findings using the Reviewer #2 method agree with our approaches. We believe we have done everything to address the issue of heating. We appreciate Reviewer #2's comments that allowed us to rigorously exclude any possibility of self-heating in our devices at low bias levels.

Reviewer #3 (Remarks to the Author):

Reviewer #4 (Remarks to the Author):

The revised manuscript NCOMMS-25-46228A by Subhajit Ghosh et al. has been improved, with respect to the original submission, and my criticisms have been adequately addressed. In particular, additional data of measurements performed on a different system (NbS3 nanowires) undergoing a similar CDW regime than (TaSe4)2I nanowires have been included in this revised version, showing that the results obtained are not just material-specific observations but are more general.

Response: We thank Reviewers #3 and #4 for stating that “criticisms have been adequately addressed.” We are glad that the significant extra work that we carried out for a second material system (which was planned as a separate paper) was appreciated by the Reviewers.

However, I do not agree with one statement of the authors (end of Page 8 of the revised version) where they assert that: “in this work, it is reported for the first time the observation of a decreasing noise level by increasing the current density”. While, I remember other works in literature reporting systems where a decrease of the noise level is observed with an increasing bias current. This is the case of disordered LBMO thin films or the case of systems undergoing weak-localization effects (<https://doi.org/10.3390/coatings11010096>). My suggestion to the authors is to take into account also these experimental results already published.

In the same review paper (<https://doi.org/10.3390/coatings11010096>), moreover, there is also a section reporting a detailed study of fluctuation mechanisms induced by charge density waves conduction in PCMO epitaxial thin films. The authors should consider these investigations by evidencing similarities or differences between their conclusions and the other ones already published.

Response: As requested, we removed the word “first time”. We modified the sentence in the text and cited the recommended paper published in the Coatings journal: Barone, C. & Pagano,

S. What Can Electric Noise Spectroscopy Tell Us on the Physics of Perovskites? Coatings 11, 96 (2021).

We do believe that our findings are significant and constitute a major advancement in the state of the art. We thank you again, Reviewers #3 and #4, for their constructive comments and suggestions.

RESPONSE TO REVIEWERS' COMMENTS

REVIEWERS' COMMENTS

Reviewer #2 (Remarks to the Author):

The main text of this last revised version has been slightly amended with additions in Supplementary. To get a final point to this tangled question of Fig.1d, it is now recognized that the data in this figure do not concern Device 1 and are just an illustration from depinning obtained from a different sample (that has to be indicated in the figure caption). Note however that the new fig1d exhibits a T dependence (non-uniform) of the threshold field different of that in the former fig1d in the original and first revised version.

Heating effects may affect measurements for any bias voltage: above threshold they are hidden and included in the non-linear variation of the resistance; below threshold they are detected from the non constant differential resistance. These effects are always present even if their consequences can have a limited effect.

Even if I feel uncomfortable with some assumptions/conclusions in this paper, I acknowledge the work done and I would not oppose the publication of this manuscript. Nevertheless, if this work meets some interest in the community, it can be refuted in future publications

RESPONSE: *We thank the Reviewer for his/her consent to publish our manuscript in Nature Communications and his/her acknowledgment of the work we accomplished.*

Figure 1 (d) in the latest resubmission corresponds to Device 1. It has been indicated in the caption. The V_t value shows a non-monotonic temperature dependence consistent with trends reported for other quasi-1D CDW systems. This has been explained in the text. We have proven that our noise measurement results and conclusions are not affected by any local heating using four different methods. The details are provided in the previous three responses.

I list below some points that can be considered by the authors; it would be worth to integrate their answers in a final text:

- All the analysis is based on the assumption that the fluctuations of the normal and collective fluctuations are independent without cross-correlation. The green line in fig.4c indicates the constant value of noise of electrons as a function of bias voltage. It might be useful to ascertain this hypothesis and beyond the two-fluid model to take into account the CDW-normal carrier interactions

RESPONSE: *Yes, the Reviewer is correct. We made a conventional assumption that the fluctuations of the condensate and normal electrons are uncorrelated. We have*

stated this in the earlier version and emphasized this in the final version. Inclusion of the interactions of the CDW condensate and normal electrons via cross-correlated terms is a separate theoretical task, which we reserve for future study. However, we are confident that the inclusion of the cross-correlations will not change the conclusions about the noise reduction and its current dependence. On the other hand, the comprehensive phenomenological model, which we developed to analyze the experimental results for CDW material systems, already constitutes a significant advancement in the state-of-the-art, which has to be reported to the community. We note that previous models focused on the noise near the depinning point, which is extrinsic by its nature. No prior models separated the three different terms contributing to noise. No prior models addressed the intrinsic noise of the sliding CDW condensate and its current dependence.

- Eq.2 gathers two different models: one described as a single particle model which implies a kink or a negative differential resistance at E_t and the other one treated as a dynamical critical phenomena. It would be necessary to be more explicit on the theoretical basis which leads to Eq. 2, beyond reference 40. To what extent of E the $(E - E_t)\zeta$ is valid?. How this equation is justified ?. Even if the fits with the experimental data look of good quality, how to understand that the exponent, α , determined by the fits is around 1 away from the $1/2$ value for the single particle model and $3/2$ for the critical phenomenon.

RESPONSE: *Eq. (2) is a phenomenological representation of the CDW contribution to the current over the experimentally accessible sliding regime. The asymptotic exponents $1/2$ and $3/2$ quoted by Grüner apply to idealized bulk limits with a single depinning field. In our nanowire geometry, neither exponent provides an adequate description of the full experimental I-V curve.*

First, *the $3/2$ exponent does not reproduce the experimentally observed change in slope at V_t , as seen in Fig. 4e,f. Capturing the kink at threshold requires a sub-linear term with an exponent below unity.*

Second, *an exponent of $1/2$ alone underestimates the current in the higher-bias region $V \gg V_t$, where the condensate contribution grows more rapidly. Modeling the curve over the full ~ 1 V bias range, therefore, requires a term with an exponent exceeding unity.*

*Since neither parameter is correct over the full bias range, we allow the two exponents in Eq. (2) to be fitted parameters, while keeping the underlying form of the theory unchanged—one term describing the initial onset of sliding, and a second term describing the steeper high-bias response. To minimize free parameters, the ratio of the two exponents is fixed such that both terms are written using a **single** generalized conductance Γ_c ; this keeps the parametrization minimal and avoids introducing two independent prefactors.*

Using the fixed theoretical exponents $1/2$ and $3/2$ together with two separate prefactors could also produce a reasonable empirical fit, but it would not change the conclusions of the noise analysis. The central result from Eq. (4), that the contribution of threshold-field fluctuations is suppressed as $1/(V - V_t)^2$ at large bias, is determined by the derivative structure of the model and is unaffected by replacing the fitted exponent $2a = 1.8$ with the theoretical value of 1.5.

Thus, the fitted exponents are best viewed as effective parameters that allow Eq. (2) to reproduce the measured nonlinear transport over the full voltage range, while the physical conclusions of the noise analysis do not depend on their precise numerical values.

We added the following text to the final manuscript.

“Eq. (2) provides a phenomenological description of the CDW contribution to the current in our nanowire devices. The two power-law terms originate from the standard CDW current expressions quoted by Grüner⁴⁰, but the exponents a and $2a$ are employed as effective exponents that capture the voltage dependence of the sliding condensate over the full experimental voltage range. We further constrain the ratio of the exponents of the two nonlinear terms to share a single generalized conductance Γ_c with fixed dimensionality, which keeps the parametrization minimal and avoids introducing additional prefactors. This yields an accurate minimal model that reproduces the full nonlinear I-V characteristics and forms the basis for the quantitative noise analysis below.”

- With the maximum voltage used, only the noise due to fluctuations in V_t is observed. That lead the authors to consider this picture as the behavior of ideal CDW sliding as a dissipation-less, noise free process. What about the ac field generated in the sliding?.

RESPONSE: We have not claimed that we achieved “dissipation-less” current transport. We stated in the first sentence of the original abstract, “While achieving dissipation-less Frohlich current of the electron–lattice condensate appears to be impossible in real materials [...]”. We claim that we have not reached the intrinsic noise of the sliding CDW condensate in $(\text{TaSe}_4)_2\text{I}$ nanowires within the examined bias range. We stated in the original abstract, “No residual minimum noise level is reached for the current of the condensate in $(\text{TaSe}_4)_2\text{I}$ nanowires.” In this sense, the CDW condensate sliding is noiseless. The noise near the depinning point is extrinsic, related to defects. We clarified this point further in the final manuscript. We do not see the need for the AC field discussion in this paper. The AC signal proves the sliding. It requires measurements in a different frequency range. We clearly proved sliding by other means.

- Concerning $(\text{TaSe}_4)_2\text{I}$ and NbS_3 the aim was to compared fully-gapped CDW materials. NbS_3 in the phase II has 3 CDW transitions at $\text{TP}_0 = 460\text{K}$, $\text{TP}_1 = 360\text{K}$ and

TP2 =155K. If the conductivity above TP0 is of hopping conductivity type, the conductivity below TP1 is very weakly T dependent - see Fig.8 in PRB95, 0356110. There are 8 chains in the unit cell and even if the structure is complicated it appears that the electron condensation appears successively on distinct chains (similarly to NbSe₃ and monoclinic and orthorhombic TaS₃). NbS₃ below TP1 is not fully gapped CDW material. The fully dielectric state is below TP2. The best compound with a unique CDW and a fully-gapped for comparison with (TaSe₄)₂I would be blue bronze (K_{0.3}MoO₃).

RESPONSE: *We thank the Reviewer for this comment, which allowed us to improve the quality of our manuscript. Our original goal (aim) for this paper was to study and report noise specifically in (TaSe₄)₂I nanowires. We did not plan to put results for various material systems into one already complicated paper, which has both experiment and theory. However, another Reviewer requested that we do a separate study of noise in another CDW material to see if we can observe a similar effect. There was also a request to compare the results with NbSe₃. We complied with both requests.*

We do realize that there are different opinions about the nature of CDW transitions in NbS₃-II. However, what is important is that the contribution of normal electrons to transport in NbS₃-II is less than in NbSe₃. The noise returns to the normal electron limit in NbSe₃ because the transport is dominated by normal carriers. The noise reduces below the normal electron limit in NbS₃ but starts to saturate. The noise reduces below the normal electron limit in (TaSe₄)₂I nanowires without signs of saturation, in the examined bias range. We refined the description of NbS₃ in the final manuscript and provided additional references. The reference L. Sneddon, "Sliding charge-density waves - dc properties," Phys. Rev. B, 29, 719 (1984) describes a sharp contrast of NbS₃ with NbSe₃ in terms of normal electron contribution at low temperature. We also added a reference recommended by the Reviewer: S.G. Zytsev, V. Ya. Pokrovskij, et al., "NbS₃: A unique quasi-one-dimensional conductor with three charge density wave transitions," Phys. Rev. B, 95, 035110 (2017).

We thank the Reviewer for the suggestion to study blue bronze (K_{0.3}MoO₃). We will attempt to investigate the noise in blue bronze and TaS₃ for the next paper.